# GenMark: An Embedded Watermarking Scheme for Generative Audio Synthesis

## Abstract

Audio watermarking provides an effective approach for tracing and protecting synthetic audio content. Traditional methods often apply watermarking as a post-processing step, which makes the watermark vulnerable to removal or degradation through signal processing or code editing. To address these issues, our paper introduces GenMark, a novel approach that embeds watermarks directly into the decoder of neural audio generation models during training. Our approach combines time-frequency perceptual losses, a mask-based localization model, and adversarial training to ensure high audio quality and watermark robustness. Experimental results on speech and music generation tasks demonstrate superior detection accuracy (TPR: 99.9% for speech, 100.0% for music). GenMark also preserves perceptual quality with less than 2% degradation in MUSHRA scores, establishing it as a strong candidate for practical and secure watermarking in generative audio systems. The replication package can be accessed at the anonymous link.[1]

## 1 Introduction

With the rapid advancement and increasing accessibility of generative audio technologies (Xiang et al., 2017; You et al., 2021; Wang et al., 2023; Borsos et al., 2023; Suno, 2023; Copet et al., 2024), concerns about the abuse of synthetic speech are growing. Modern speech synthesis models like deepfake technology (Shaaban et al., 2023) enable voice cloning that could manipulate public discourse (News, 2024), damage individual reputations (Findlay, 2025), or compromise national security (Canadian Security Intelligence Service, 2023). These risks highlight the critical need for effective detection tools and traceability measures to verify the authenticity of synthetic audio and enforce accountability.

In such cases, audio watermarking serves as an effective solution by embedding imperceptible identifiers to trace model-generated audio, which effectively prevents malicious users' misuse of synthetic audio. Current mainstream audio watermarking methods embed watermarks directly into audio signals. In audio generation scenarios, it requires first generating audio by a generation model and then embedding a watermark into the generated audio. However, the post-processing watermarking strategy poses a serious security risk: malicious users can take control of the watermarking embedding process. By circumventing the watermark embedding stage, they are able to produce unwatermarked audio and exploit it in illicit scenarios (Wen et al., 2025; O'Reilly et al., 2025). This poses a huge challenge to the regulation of synthetic audio. Moreover, audio generation poses unique challenges for watermarking, such as dealing with intricate frequency patterns and ensuring that the watermark stays reliable without affecting audio quality (Salah et al., 2025). These difficulties make it hard to embed robust and imperceptible watermarks.

To address these issues, we propose GenMark, a novel in-process injection watermark method that embeds the watermark during the audio generation process. GenMark improves traditional post-generation watermarking by directly generating audio with embedded watermarks. Unlike traditional post-generation watermarking methods, GenMark allows direct generation of audio with embedded watermarks. This prevents malicious attackers from manipulating the watermarking process and ensures reliable regulation of synthetic audio. Instead of modifying the entire generation pipeline, we focus only on the decoder, which converts tokens into audio samples. This choice enables efficient

---

[1] https://anonymous.4open.science/r/Gen-Mark-1F27

integration while maintaining generation quality. `GenMark` leverages joint time-frequency losses to improve perceptual audio quality and incorporates a mask model to enhance watermark robustness and location accuracy. In addition, it adopts GAN-based training to enhance the imperceptibility of the watermark. As a result, the generated waveforms inherently encode persistent and identifiable watermark signatures, regardless of input prompts or decoding parameters.

We evaluate `GenMark` using four state-of-the-art watermarking models, WavMark (Chen et al., 2024), AudioSeal (San Roman et al., 2024),and SilentCipher (Singh et al., 2024) on both speech and music generation tasks. In terms of audio quality, `GenMark` consistently achieves lower Frechet Audio Distance (FAD) and Kullback-Leibler Divergence (KLD) scores across multiple datasets, indicating minimal perceptual and distributional distortion. It also maintains strong semantic alignment, outperforming baselines on the CLAP metric. For detection, we report *TPR*, *FPR*, and *decode accuracy*. Our method outperforms baselines in both detection and watermark recovery. To evaluate robustness, we subject watermarked audio to 12 common audio transformations and adversarial attacks, comparing the decoding error rates with those of WavMark, AudioSeal, and SilentCipher. Besides, subjective MUSHRA evaluations further confirm that `GenMark` preserves perceptual quality and the ablation studies show that each component of `GenMark` contributes to the balance between fidelity, robustness, and detection precision. We summarize contributions as follows:

- We propose `GenMark`, a novel framework that embeds inaudible watermarks directly into generative audio models during training.
- `GenMark` introduces a multi-scale discriminator and a mask model to improve audio quality and watermark robustness.
- Experiments show near-perfect detection rates (TPR: 99.9% for *Bark*, 100.0% for *MusicGen*) with FPR $\leq$0.1%. `GenMark` maintains low decoding error rates under 12 distortions and less than 2% perceptual degradation in MUSHRA tests, outperforming state-of-the-art baselines.

## 2 PRELIMINARIES

### 2.1 AUDIO GENERATION

The current neural audio generation systems follow a hierarchical processing pipeline. Multi-modal inputs—such as text or speech prompts—are first encoded into discrete acoustic tokens through cascaded transformer layers (Vaswani et al., 2017). These tokens serve as high-level latent representations of the target audio. To synthesize natural-sounding waveforms, the tokens are then passed through spectral enhancement modules, including neural vocoders (Kong et al., 2020) and differentiable signal processing components (Engel et al., 2020). Finally, the decoder transforms the processed acoustic tokens and synthesizes them into the final audio waves.

### 2.2 LOSS BALANCER

In multi-objective training settings, gradients from different loss terms can vary significantly in scale. This imbalance may lead to unstable optimization and make the effect of each loss weight $\lambda$ hard to interpret. To address this, we adopt loss balancers inspired by EnCodec (Défossez et al., 2022), which dynamically rescales gradient contributions based on their recent magnitude.

For each loss $\mathcal{L}_i$, we compute its gradient $g_i = \frac{\partial \mathcal{L}}{\partial \hat{x}}$ and track the exponential moving average of its norm $\|g_i\|_2^\beta$. Then, the rescaled gradient is,

$$\tilde{g}_i = \frac{R \cdot \lambda_i}{\sum_j \lambda_j} \cdot \frac{g_i}{\|g_i\|_2^\beta}. \tag{1}$$

The final gradient used for backpropagation is $\sum_i \tilde{g}_i$, instead of the original $\sum_i \lambda_i g_i$, which helps stabilize training. The $R$ is a reference gradient scale, and the $\beta$ is a decay rate.

### 2.3 WATERMARK

Watermarking embeds extra information (a payload) into an audio signal in a way that is ideally imperceptible to human listeners, yet still reliably recoverable by a detector under various distortions. The key design goals are: Audio Quality, Robustness, Efficiency and Detection Reliability.

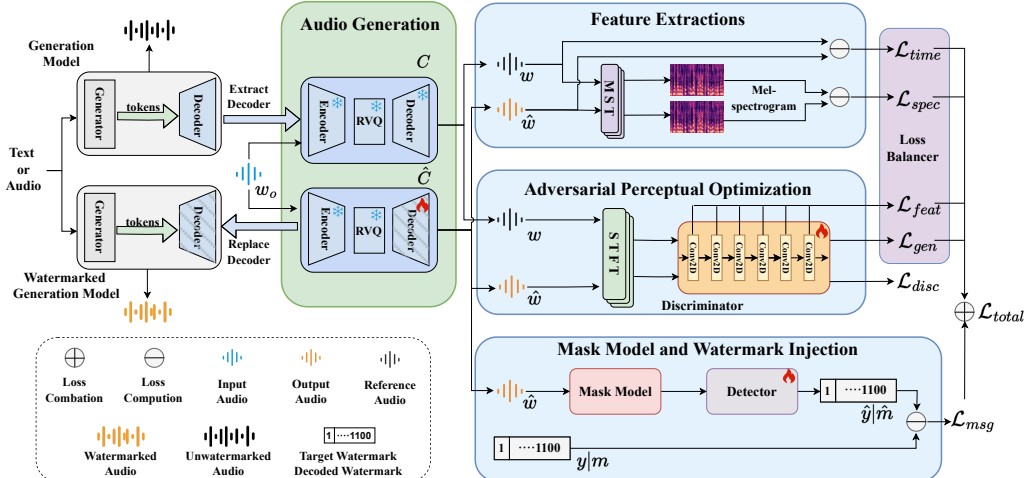

Figure 1: Overview of the training pipeline. First, a frozen reference codec $C$ generates clean, unwatermarked audio, while a trainable codec $\hat{C}$ updates only its decoder to embed watermarks. Next, losses between the clean and watermarked outputs are computed to maintain audio quality, combining standard perceptual and adversarial terms; the discriminator is optimized with $\mathcal{L}_d$, and the message-loss weight $\mathcal{L}_m$ is tuned separately for effective watermark embedding. To further improve robustness, a mask model is applied, and finally a dedicated decoder network extracts the embedded watermark from the output. Pseudocode for the complete procedure appears in Appendix B.

In classical audio watermarking, methods often operate in transform domains (e.g. DCT, DWT, spread-spectrum) or use echo/phase manipulation. With the rise of neural methods, watermarking is now commonly integrated into the embedding–detection pipeline via learnable networks. The embedder (or "encoder") inserts the watermark signal into an audio representation, while the detector (or "decoder") extracts it.

## 3 METHODOLOGY

GenMark employs gradient steganography to embed watermark signals directly into the generative process by optimizing the decoder component of the model. Instead of modifying the entire generation pipeline—which is often large and difficult to fine-tune—we target the decoder, the final stage responsible for converting discrete token sequences into audio waveforms. This position makes it particularly suitable for learning robust watermark patterns. By training the decoder to produce watermarked audio without compromising perceptual quality, we enable direct integration of watermarking into the model. Once trained, the decoder can be seamlessly substituted for the original one, enabling watermark embedding without modifying the rest of the generation pipeline.

### 3.1 TRAIN PIPELINE

**Overview.** To embed watermark information $m$ into the parameters of the decoder, we guide the decoder's optimization using a joint loss, which includes perceptual loss ($\mathcal{L}_{\text{time}}, \mathcal{L}_{\text{spec}}$), adversarial loss ($\mathcal{L}_{\text{gen}}, \mathcal{L}_{\text{disc}}, \mathcal{L}_{\text{feat}}$), and decoding loss ($\mathcal{L}_{\text{msg}}$). We balance these objectives during training by scaling their gradient contributions using the Loss Balancer 2.2. The full training pipeline consists of four stages, as illustrated in Figure 1.

**Audio generation.** Firstly, we extract the compression model (also known as a codec, such as EnCodec) from the audio generation model (e.g., *Bark*). The codec $\hat{C}$ consists of an encoder, a quantizer, and a decoder, which together map raw waveforms to discrete tokens and reconstruct audio from them. During optimization, we freeze the encoder and quantizer of $\hat{C}$ and only update its decoder, which converts tokens into waveforms. This setup enables efficient watermark embedding by modifying only the decoder.

Given an input audio signal $w_o \in \mathbb{R}^T$, the codec $\hat{C}$ generates a watermarked version $\hat{w} \in \mathbb{R}^T$. For reference, we use an untrained copy of the same codec, denoted $C$, to reconstruct a non-watermarked

version $w$. Then, we simultaneously optimize for two objectives: enabling reliable watermark decoding from $\hat{w}$, and minimizing the difference between $w$ and $\hat{w}$ to preserve audio quality.

**Feature extractions.** To preserve perceptual audio quality, we compute time-domain $\mathcal{L}_{\text{time}}$ and frequency-domain $\mathcal{L}_{\text{spec}}$ losses between $w$ and $\hat{w}$. The time-domain loss constrains waveform-level distortions, promoting time-domain alignment. The frequency-domain loss is calculated using multi-scale Mel spectrograms, which are widely used to reflect human auditory perception and capture perceptual differences across resolutions (Kong et al., 2020; You et al., 2021). This hybrid loss strategy has been shown to be effective in maintaining perceptual fidelity in neural audio synthesis, such as (Tan et al., 2024; Zhang et al., 2019; Yamamoto et al., 2020), as well as in compression tasks (Défossez et al., 2022; Zeghidour et al., 2021).

**Adversarial Perceptual Optimization.** To improve audio quality and reduce perceptual artifacts, we adopt adversarial training following prior works (Défossez et al., 2022). As illustrated in Figure 1, the decoder of the codec serves as the generator, producing watermarked audio $\hat{w}$, while a lightweight multi-scale discriminator distinguishes $\hat{w}$ from the reference audio $w$. The adversarial loss for the generator is $\mathcal{L}_{\text{gen}}$ and for the discriminator is $\mathcal{L}_{\text{dic}}$. Similarly to previous work (You et al., 2021; Kong et al., 2020), we also incorporate a feature-matching loss $\mathcal{L}_{\text{feat}}$ for the generator.

**Maks Model and Watermark Injection.** The watermarked audio $\hat{w}$ is further processed by a mask model $M$ (in Section 3.4) designed to enhance robustness and enable fine-grained watermark localization. The model comprises two components: a Localization Refinement Module, and a Robustness Enhancement Module module. They ensures the watermark remains detectable under common audio modifications while reducing false positives.

After that, the audio is fed to the watermark detector $D_{det}$, which outputs $D_{det}(\hat{w}) \in [0, 1]^{18 \times T}$. The first two dimensions of $D_{det}(\hat{w})$ represent the frame-level probabilities of watermark presence, while the remaining 16 dimensions correspond to the decoded 16-bit watermark sequence. This prediction is then compared with the target watermark message $m$, and the discrepancy is used to compute the decoding loss $\mathcal{L}_{\text{msg}}$, guiding the model to embed the watermark into the audio. The architecture details of $D_{det}$ are provided in the Appendix G.

## 3.2 FEATURE EXTRACTIONS.

Although the primary objective is to embed watermark signals into the audio, it is crucial that the perceptual quality of the output remains unaffected. To ensure this, the audio fidelity loss incorporates complementary constraints across both time and frequency domains, informed by principles of human auditory perception (Xiang et al., 2017),

$$\mathcal{L}_{\text{time}} = \|w - \hat{w}\|_1. \tag{2}$$

Eq. (2) promotes robust waveform similarity while remaining minor phase variations that have minimal perceptual impact (Engel et al., 2020).

However, as human auditory perception varies in sensitivity across different frequency ranges, optimization in the time domain alone may not suffice to achieve high-quality audio perception. To address this, we introduce a Multi-scale Mel Spectrogram Loss (Gritsenko et al., 2020), which constrains the spectral characteristics (frequency domain feature) of the generated audio. Eq. ( 3) uses a multi-resolution Mel-spectrogram analysis with window sizes set $\mathcal{H} = \{32, 64, 128, 256, 512, 1024\}$. And $S_h(\cdot)$ denotes the function of the Mel-spectrogram using a fixed window size $h$:

$$\mathcal{L}_{\text{spec}} = \sum_{h \in \mathcal{H}} \sum_{i=1,2} \left[ \|S_h(w) - S_h(\hat{w})\|_i \right]. \tag{3}$$

The combination of absolute difference ($\ell_1$) and squared difference ($\ell_2$) formulation balances spectral magnitude alignment with overall distribution consistency (Gritsenko et al., 2020), reducing the over-smoothing effects often observed in pure $\ell_2$ optimization (Kong et al., 2020).

## 3.3 ADVERSARIAL PERCEPTUAL OPTIMIZATION.

Although feature-based losses help maintain the overall perceptual quality of audio, they may not fully capture subtle distortions or unnatural details that can still affect the quality of audio. To further

enhance perceptual realism and improve watermark robustness, we adopt an adversarial training strategy using multi-scale spectral discriminators, inspired by prior work on neural vocoders and audio synthesis (Défossez et al., 2022; You et al., 2021; Kong et al., 2020).

The discriminator architecture follows a five-layer dilated convolutional design with dilation rates $[1, 2, 4]$, weight normalization, and LeakyReLU activations ($\alpha = 0.2$) for stable convergence. It processes the input across multiple spectral resolutions in parallel, using STFTs with FFT sizes $\{512, 1024, 2048\}$ and corresponding window lengths $\{128, 256, 512\}$. This multi-scale structure enables the discriminator to capture both fine- and coarse-grained spectral artifacts, making it a strong perceptual sensitivity.

**Generator Objective.** The generator $G$ is trained to generate watermarked audio that is perceptually indistinguishable from original signals:

$$\mathcal{L}_{\text{gen}} = \mathbb{E}_{\hat{w}} \left[ \mathbb{E}_{k \in \mathcal{K}} \| 1 - D_k(\hat{w}) \|_1 \right], \tag{4}$$

where $\mathcal{K}$ represents the STFT window size set set, and $D_k(\cdot)$ is the discriminator output.

In addition, inspired by prior work (Kumar et al., 2019b; Kong et al., 2020; You et al., 2021; Défossez et al., 2022), we include a feature-matching loss $\mathcal{L}_{\text{feat}}$ encourages the generator to produce internal representations that closely resemble those extracted from real audio by the discriminator:

$$\mathcal{L}_{\text{feat}} = \mathbb{E}_{l \in \mathcal{S}, k \in \mathcal{K}} \left[ \frac{\| D_k^l(w) - D_k^l(\hat{w}) \|_1}{\mathbb{E}[D_k^l(w)] + \epsilon} \right], \tag{5}$$

where $\mathcal{S}$ denotes the set of discriminator layers, and $D_k^l$ represents the output of the $l$-th layer of the discriminator corresponding to an STFT window size $k$. The term $\epsilon = 10^{-6}$ is introduced to prevent division by zero.

**Discriminator Objective.** The discriminator $D$ is optimized to differentiate between real and watermarked audio signals:

$$\mathcal{L}_{\text{dic}} = \mathbb{E}_w \left[ \mathbb{E}_{k \in \mathcal{K}} \| 1 - D_k(w) \|_1 \right] + \mathbb{E}_{\hat{w}} \left[ \mathbb{E}_{k \in \mathcal{K}} \| D_k(\hat{w}) \|_1 \right].$$

By leveraging adversarial, feature-matching, and detector losses with multi-scale discriminators, the model improves the perceptual realism of audio while enhancing the robustness of the watermark.

### 3.4 MAKS MODEL AND WATERMARK INJECTION

#### 3.4.1 MAKS MODEL

In order to reduce the false positive rate, improve localization accuracy, and enhance watermark robustness, we additionally include an enhanced mask module, which exposes the decoder to a variety of masking patterns during training, enabling it to better distinguish true watermark signals, improve its resilience to common audio attacks.

**(1) Localization Refinement Module**: To reduce false positives and improve spatial precision, we introduce two training strategies: (a) part of watermarked segments are replaced with alternative watermark patterns to prevent overfitting; (b) within each audio, $K$ regions are randomly selected and partially replaced with clean, unrelated, or silent content. These perturbations force the decoder to learn precise localization and improve extraction accuracy by distinguishing true watermark regions from distractors. The parameter settings are elaborated in Appendix C.

**(2) Robustness Enhancement Module**: To improve the watermark's resilience to signal processing attacks, we develop a sequential transformation pipeline that applies nine fundamental audio operations in carefully calibrated proportions, including frequency filtering, resampling, dynamic range adjustment, echo effects, noise addition, and waveform smoothing. This transformation is commonly used in watermark removal attacks and watermark robustness enhancement (Kirovski & Malvar, 2003; Li et al., 2024). By simulating these attacks during training, the decoder learns to maintain watermark fidelity. The probability and parameters of each operation (e.g., frequency thresholds for filtering, signal strength for noise addition) are carefully optimized, as outlined in Appendix D.

### 3.4.2 WATERMARK INJECTION

To ensure stable and accurate watermark recovery, we define a message loss that guides the model to retain the correct message content during decoding. It consists of two core components:

$$\begin{cases} \mathcal{L}_{\text{det}} = \frac{1}{T} \sum_{t=1}^{T} \left[\text{BCE}\left(y_t, \hat{y}_t\right)\right] \\ \mathcal{L}_{\text{payload}} = \frac{1}{T} \sum_{t=1}^{T} \left[\text{BCE}\left(m_t, \hat{m}_t\right)\right], \end{cases} \tag{6}$$

where $y_t \in \{0, 1\}$ denotes the presence of a watermark in frame $t$, and $m_t \in \{0, 1\}^{16}$ corresponds to the ground-truth 16-bit message. The overall watermark loss function is formulated as:

$$\mathcal{L}_{\text{msg}} = \lambda_{\text{det}} \mathcal{L}_{\text{det}} + \lambda_{\text{payload}} \mathcal{L}_{\text{payload}}, \tag{7}$$

where $\lambda_{\text{det}}$ and $\lambda_{\text{payload}}$ balance the importance of detection accuracy and payload reconstruction.

As described in the training pipeline, the decoder $D$ receives the masked audio output from the Mask Model and produces a tensor $D(\hat{w}) \in [0, 1]^{18 \times T}$, where each of the $T$ frames contains detection and decoding information.

## 4 EXPERIMENTS SETTING

**Models and Datasets.** We use two state-of-the-art generative models, *Bark* (Suno, 2023) for speech synthesis and *MusicGen* (Copet et al., 2024) for musical audio generation, to insert a watermark. Training and evaluation are conducted on AudioSet (Gemmeke et al., 2017) and CommonVoice (Foundation, 2020) dataset, ensuring diverse coverage of both general acoustic environments and multilingual speech. Since *Bark* requires textual prompts as input, we additionally incorporate several text-based datasets as test cases to evaluate watermarking performance: HarvardSentences (on Subjective Measurements, 1969) and LJSpeech (Ito, 2017). These setups enable a comprehensive assessment of our watermarking method across speech and non-speech domains.

**Training Configuration.** All models are trained on an NVIDIA RTX 3090 GPU with an initial learning rate of $1 \times 10^{-4}$, which is gradually decreased for stable convergence. Batch sizes are set to 24 for *Bark* and 16 for MusicGen, reflecting their respective computational demands. To accommodate the inherent sampling preferences of these models, *Bark* is trained at 24 kHz, while MusicGen is trained at 32 kHz. We balance our multi-objective loss using the balancer with $\lambda_{\text{time}} = 1$, $\lambda_{\text{freq}} = 6$, $\lambda_{\text{gen}} = 9$, $\lambda_{\text{feat}} = 9$, $\lambda_{\text{msg}} = 10$ . The discriminator updates once every two epochs, allowing the generator sufficient adaptation time and ensuring more stable adversarial training.

**Baselines.** GenMark is compared with several baselines: (1) AudioSeal (San Roman et al., 2024), (2) Wavmark (Chen et al., 2024), and (3) SilentCipher (Singh et al., 2024). These methods are recognized for their effectiveness in audio watermarking, and together, they provide a strong benchmark for evaluating imperceptibility, robustness, and decoding accuracy across various audio conditions.

## 5 EXPERIMENTS RESULT

We evaluate GenMark in four key aspects: audio quality, detection accuracy, robustness, and human perception. Specifically, we assess whether watermarking affects audio quality, measure detection performance across different models, test robustness under common audio perturbations, and conduct a subjective listening study to understand the impact on human listeners. In addition, we conduct ablation studies to validate the effectiveness of key components and analyse the Mel spectrograms of watermarks.

### 5.1 QUALITY OF AUDIO

To explore GenMark 's capability to preserve perceptual and semantic quality in synthetic audio, we evaluate the similarity between the generated watermarked audio and original audio samples based on distributional (SISNR, KLD), perceptual (FAD, VISQOL), and semantic metrics (CLAP). A detailed introduction to the metrics is in the appendix E.

*Distributional Quality.* In practice, high SI-SNR is indeed not necessarily correlated with good perceptual quality (San Roman et al., 2024). GenMark has not specifically optimized for SISNR. But it consistently achieves better performance in other metrics(KLD, FAD, VISQOL and CLAP)

Table 1: Model comparison under perceptual (FAD/VISQOL), distributional (KLD/SI-SNR), and semantic (CLAP) metrics.

| Model | HarvardSentence | | | | | LJSpeech | | | | |
|---|---|---|---|---|---|---|---|---|---|---|
| | SISNR | KLD | FAD | VISQOL | CLAP | SISNR | KLD | FAD | VISQOL | CLAP |
| AudioSeal | 24.80 | 0.2029 | 0.4533 | 4.6006 | 9.28 | 25.23 | 0.1727 | 0.1976 | 4.5919 | 8.67 |
| WavMark | 27.86 | 0.1526 | 1.6092 | 3.4278 | 9.15 | 27.48 | 0.1641 | 1.5716 | 3.4460 | 9.49 |
| SilentCipher | **48.26** | 0.1370 | 0.2936 | 4.5688 | 9.39 | **39.00** | 0.1375 | 0.1794 | 4.4247 | 8.81 |
| Ours | 24.25 | **0.1321** | **0.0615** | **4.7172** | 9.28 | 25.99 | **0.1364** | **0.0227** | **4.7169** | **8.11** |

*Perceptual Quality.* We use FAD(the lower is better) and VISQOL (the higher is better) as our Perceptual metrics.Conceptually, FAD captures corpus-level perceptual shift, whereas VISQOL targets utterance-level signal fidelity; GenMark performs strongly on both, attaining the lowest FAD and highest VISQOL on HarvardSentence (0.0615 / 4.7172) and LJSpeech (0.0227 / 4.7169), indicating preserved naturalness at both levels.

*Semantic Consistency.* GenMark achieves the best CLAP scores on LJSpeech (8.11), outperforming all baselines. On HarvardSentence, it is slightly behind WavMark (9.28 vs. 9.15), but still ahead of other methods. These results demonstrate that GenMark consistently preserves semantic alignment while embedding watermark signals.

From these results we learn that high SI-SNR alone does not guarantee perceptual quality — GenMark instead achieves state-of-the-art perceptual naturalness while maintaining strong semantic consistency.

## 5.2 DETECTION ACCURACY

To assess the efficacy of our watermarking technique, we conducted comprehensive detection experiments using two prominent generative audio models: *Bark* and *MusicGen*. *Bark* is designed for high-quality speech synthesis, whereas *MusicGen* is tailored for generating musical audio. We generate and analyze 10,000 audio samples per method for each model to ensure statistically reliable results. Detection performance is measured using TPR and FPR, as presented in Table 2.

As shown in Table 2, GenMark has strong detection performance. For the *Bark* model, our method achieves a TPR of 99.9% with zero false positives, while attaining perfect detection (100.0% TPR) on *MusicGen* with a minimal FPR of 0.1%. Although AudioSeal also achieves high TPRs, especially on *Bark*, it shows a noticeable drop in accuracy on *MusicGen*. In contrast, our method maintains balanced performance across both domains. WavMark exhibits similar accuracy to our method on *Bark* but falls short in TPR on *MusicGen*. SilentCipher's performance is less stable overall, with high false positives observed in both settings.

Table 2: Detection results for *Bark*, *Musicgan* with TPR, FPR and Decode Accurate (%).

| Model | Bark | | | Musicgan | | |
|---|---|---|---|---|---|---|
| | TPR | FPR | Acc | TPR | FPR | Acc |
| Audioseal | 100.0 | 0.0 | 95.4 | 100.0 | 0.1 | 73.3 |
| Wavmark | 99.8 | 0.0 | 99.8 | 95.2 | 0.1 | 94.4 |
| SilentCipher | 92.4 | 31.4 | 96.6 | 98.2 | 39.6 | 97.8 |
| Ours | 99.9 | 0.0 | 99.8 | 100.0 | 0.1 | 94.3 |

As the reaults show, Our method delivers consistently robust and precise detection across both speech and music domains, outperforming prior approaches in stability and reliability.

## 5.3 ROBUSTNESS OF WATERMARK

We assess GenMark 's robustness to real-world perturbations using *Bark* as the generative backbone and 12 audio transformations. Robustness is measured by the decoding error rate (lower is better) for each transformation. As shown in Table 3, GenMark attains the lowest error on 9 of 12 transformations (ties counted), excelling on echo, ducking, speed change, bandpass, boost, and both pink/white noise, where errors are typically $\leq 3\%$. Even under the difficult lowpass setting—where competing methods often fail—our error remains about $51\%$.

Although *Highpass*, *Encodec*, and *Black-box* transformations degrade performance more noticeably, GenMark remains strongest overall (mean error 12.69% vs. WavMark 43.14%, SilentCipher 54.84%, AudioSeal 54.92%). The drop under *Highpass* is expected: to preserve perceptual quality we intentionally concentrate watermark energy in mid–low frequencies, so aggressive high-pass filtering removes a larger fraction of the embedded signal. For *Encodec* and *Black-box*, we did not

Table 3: Decoding Error Rates (%) under different audio transformations.

| Model | Audio Transformations (Decoding Error Rates %) | | | | | | | | | | | | Total |
|---|---|---|---|---|---|---|---|---|---|---|---|---|---|
| | Bandpass | Highpass | Lowpass | Speed | Boost | Duck | Echo | Pink | White | Encodec | White-box | Black-box | |
| AudioSeal | 92.08 | 100.00 | 100.00 | 99.85 | 29.24 | 95.63 | 15.61 | 23.68 | 54.72 | 42.17 | 6.03 | **0.03** | 54.92 |
| WavMark | 0.21 | **0.13** | 100.00 | 97.57 | 7.83 | 4.89 | 3.95 | 79.23 | 99.72 | **6.25** | 17.95 | 100.00 | 43.14 |
| SilentCipher | 34.58 | 43.26 | 97.70 | 99.16 | 6.66 | 6.78 | 79.79 | 100.00 | 100.00 | 13.75 | 28.69 | 47.73 | 54.84 |
| Ours | **0.05** | 68.57 | **50.97** | **1.36** | **1.24** | **0.17** | **0.17** | **1.62** | **3.12** | 8.53 | **3.57** | 12.86 | **12.69** |

include codec-specific or pipeline-aware adversarial optimization during training, so quantization and unknown post-processing introduce an unaddressed domain gap.

From these experiments we learn that the mask model (especially the robustness enhancement module) can substantially improve robustness to perturbations — as also evidenced in our ablation studies in Section 5.6.

## 5.4 Usable Study

To assess perceptual audio quality from a human perception perspective, we perform a subjective evaluation using the standardized MUSHRA (MUltiple Stimuli with Hidden Reference and Anchor) protocol (ITU-T, 2015), a well-established methodology widely adopted for audio quality benchmarking. We invite 20 audio experts to evaluate 20 audio groups, each corresponding to a distinct prompt. For every prompt, one sample was randomly selected from 100 *Bark*-generated clips. Each group includes the following: (1) three types of watermarked audio samples (GenMark, AudioSeal, WavMark); (2) one clean reference; and (3) two anchor sig-

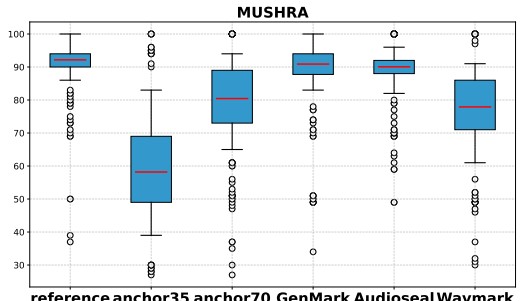

Figure 2: Distribution of MUSHRA scores for watermarked audio.

nals, namely Anchor35 (filtered at 3.5 kHz) and Anchor70 (filtered at 7 kHz). Participants rate each sample on a scale of 0–100, with anchors and references used to guide their judgments. Details are provided in Appendix H.

As presented in Figure 2, our proposed method achievess the highest MUSHRA score (**90.89**), closely followed by AudioSeal (**90.06**), with WavMark lagging at **77.90**. For comparison, the clean reference audio achieves a MUSHRA score of **92.17**, while the Anchor70 and Anchor35 conditions score **80.44** and **58.18**, respectively.

These subjective evaluations highlight that GenMark can embed robust watermarks while preserving perceptual quality nearly indistinguishable from clean audio - less than 2% perceptual degradation in MUSHRA test.

## 5.5 Mel-spectrogram Analyse

We analyze a representative utterance and compute Mel spectrograms for the original signal and each watermark component (Fig. 3). Temporally, the strength of the watermark follows the shape of the original audio, but across frequencies, each method shows its own differences.

GenMark and SilentCipher exhibit the same frequency patterns as the original signal, with most watermark energy concentrated at or below $4\,\mathrm{kHz}$. SilentCipher, in particular, aligns almost perfectly with the original distribution, which coincides with the highest SI-SNR among the compared methods. By contrast, AudioSeal concentrates watermark energy in lower bands (below $2\,\mathrm{kHz}$). WavMark spreads the watermark across the whole spectrum and leans toward the high end (above 8 kHz). This can mask or remove high-frequency details. In our MUSHRA tests, listeners noticed a clear loss of high-band detail in audio processed by WavMark.

Consequently, embedding watermark energy in frequency bands aligned with the original audio's dominant energy produces less perceptual degradation — deviations from that alignment tend to introduce noticeable quality loss.

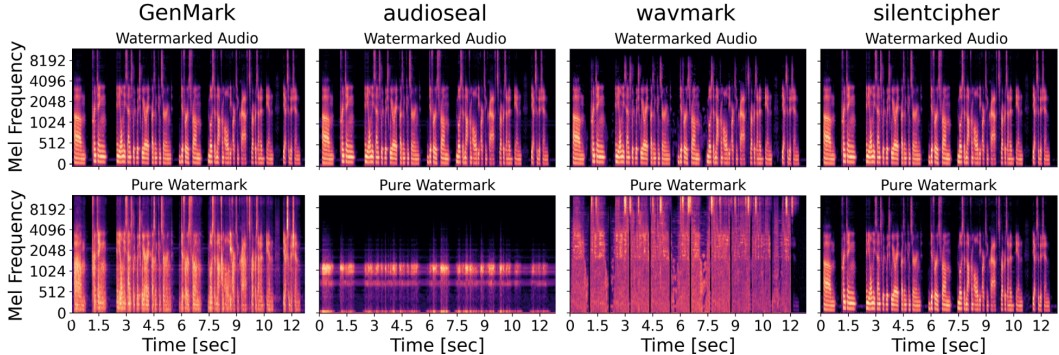

Figure 3: Mel-spectrograms of (top) the watermarked audio and (bottom) the isolated watermark, obtained by subtracting the original signal from the watermarked signal.

## 5.6 ABLATION STUDY

To understand the contribution of each component, we conduct an ablation study focusing on three core modules: (1) adversarial perceptual optimization, (2) the robustness enhancement module, and (3) the localization refinement module. For each variant, we remove or disable one of the components and evaluate performance on key metrics, including detection (TPR, FPR, Acc), perceptual quality (FAD), and robustness

Table 4: Ablation study of `GenMark` evaluating the effect of each component.

| Variant | TPR↑ | FPR↓ | Acc↑ | FAD↓ | DER↓ |
|---|---|---|---|---|---|
| *WithoutAdv* | 99.8 | 0.0 | 99.8 | 0.1493 | 10.92 |
| *WithoutRobust* | 99.9 | 0.0 | 99.9 | 0.0607 | 46.22 |
| *WithoutLoc* | 100.0 | 35.7 | 100.0 | 0.0595 | 6.43 |
| Full Model | **99.9** | **0.0** | **99.8** | **0.0615** | **11.55** |

(average decode error rate under transformations), as shown in Table 4. For this table, we observe that removing adversarial training (*WithoutAdv*) results in a drop in perceptual quality, as indicated by the increase in FAD from 0.0615 (full model) to 0.1493. Disabling the robustness enhancement module (*WithoutRobust*) has the most significant effect on robustness, with the average decode error rate (DER) surging from 11.55% to 46.22%. Removing the localization refinement module (*WithoutLoc*) improves robustness but at the cost of a substantial increase in FPR, highlighting its importance in maintaining detection precision.

## 6 RELATED WORK

Traditional audio watermarking techniques (Cvejic & Seppanen, 2004; Anderson, 1996) typically embed watermarks by manipulating information in the time or frequency domains (Cox et al., 1997; Xiang et al., 2018; Su et al., 2018; Liu et al., 2019). These methods depend on manually crafted heuristic rules and specialized domain expertise to guide their design and implementation. Simultaneously achieving a high imperceptibility, capacity, and robustness watermark across diverse audio types remains a significant challenge.

With advancements in deep learning, the ability to automatically learn watermark embedding and extraction techniques has simplified the design of watermarking methods (Tai & Mansour, 2019; Pavlović et al., 2022). In particular, current deep learning-based watermarking techniques generally follow an Encoder-Decoder structure (Qu et al., 2023; Ren et al., 2023; Chen et al., 2024; San Roman et al., 2024), where the encoder generates watermarked audio, and the decoder extracts the information from the watermarked audio. The entire model is trained in an end-to-end manner, enabling it to automatically learn the watermark embedding and extraction processes.

## 7 CONCLUSION

This work introduces `GenMark`, a robust and efficient method for embedding traceable, imperceptible watermarks directly into generative audio models. By integrating watermark objectives directly into the generation model, `GenMark` addresses the vulnerabilities of traditional post-generation watermarking. Extensive evaluation across speech and music generation confirms that `GenMark` offers superior detection accuracy, resilience to a wide array of audio attacks, and negligible perceptual degradation. These results establish `GenMark` as a strong tool for protecting audio synthesis systems.

## LIMITATIONS

While `GenMark` demonstrates strong performance across multiple generative audio tasks, it requires model-specific integration during training. Since the watermark is embedded directly into the decoder, each generative model (e.g., Bark, MusicGen) must be individually fine-tuned with `GenMark`.

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

# A EXTENDED RELATED WORK

## A.1 SECURITY AND MISUSE IN GENERATIVE MODELS

The rapid advancement of generative models across text (Brown et al., 2020b; Touvron et al., 2023), image (Rombach et al., 2022), and audio (Kreuk et al., 2022) domains has brought remarkable synthesis quality and expressiveness. However, with this growth comes increasing concern over misuse. Recent work has shown that generative pipelines can be tampered with or exploited, such as backdoor injection in offline reinforcement learning datasets (Gong et al., 2024), data poisoning in large language models (Carlini et al., 2023), and output evasion in diffusion models (Xu et al., 2023). These studies highlight the importance of security-aware generative model design, especially in ensuring traceability and tamper resistance.

In the audio domain, the risk is amplified by the realism of synthetic speech. Voice cloning and TTS systems have been used for impersonation, misinformation (News, 2024), and fraud (wes, 2020). Watermarking has emerged as a defense strategy (Mou et al., 2023; Chen et al., 2023; Singh et al., 2024), yet most approaches apply watermarks after generation, leaving them vulnerable to removal or circumvention. Our work addresses this gap by embedding watermarks directly during training, offering stronger protection against post-generation manipulation.

## A.2 COMPRESSION MODEL

SoundStream (Zeghidour et al., 2021) and EnCodec (Défossez et al., 2022) are neural audio codecs designed for high-fidelity audio compression and reconstruction. SoundStream introduces a fully learnable end-to-end framework using residual vector quantization, while EnCodec builds upon this design with improved scalability and audio quality through hierarchical quantization and adversarial training. These models pioneer neural audio compression through self-supervised learning and hierarchical quantization. Unlike traditional handcrafted feature methods, these approaches efficiently encode high-dimensional audio into discrete tokens, retaining semantic information.

This tokenization framework empowers Transformer-based systems (e.g., *Bark* (Suno, 2023), Music-GAN (Copet et al., 2024), AudioLM (Borsos et al., 2023)) to perform cross-modal audio generation from text prompts and context-aware audio continuation. By integrating audio compression with language model architectures, these methods improve efficiency and versatility in generative AI, facilitating a wide range of multimodal synthesis applications.

## A.3 ATTACKS ON AUDIO WATERMARKING SYSTEMS

While audio watermarking enables traceability of generated content, ensuring robustness under adversarial or lossy conditions remains a major challenge. Watermarks are often vulnerable to signal manipulations such as compression, noise injection, cropping, pitch shifting, or time-stretching (Cox et al., 2007; Arnold et al., 2003). Attackers can intentionally apply these distortions to remove or degrade the watermark information without significantly affecting audio perceptual quality.

Classical attack strategies include re-encoding, filtering, jittering, or frequency band removal (Wang et al., 2004). Recent works even explore adversarial perturbations designed specifically to confuse watermark extractors (Wu et al., 2022). Therefore, the evaluation of watermark robustness must consider both standard degradations (e.g., MP3 compression, resampling) and targeted attacks (e.g., masking, inversion, audio remix).

In our experiments, we systematically test GenMark under 11 widely used audio transformations and adversarial manipulations to benchmark its resistance. Our method demonstrates lower decoding error rates compared to WavMark, AudioSeal, and SilentCipher, showing enhanced watermark durability under attack.

## A.4 AUDIO GENERATION

Currently, audio generation has evolved significantly through deep learning. For instance, autoregressive models such as WaveNet (Van Den Oord et al., 2016) greatly improve audio quality, whereas

---

**Algorithm 1** Training pipeline for `GenMark`

---

**Require:** Dataset $\mathcal{D} = \{w_o\}$, message sampler $\text{SAMPLEMSG}(\cdot)$, reference codec $C$ (frozen), trainable codec $\hat{C}$ (frozen encoder & quantizer; trainable decoder with params $\theta$), mask model $M$, watermark detector $D_{\text{det}}$ (params $\phi$), multi-scale discriminator $D$ (params $\psi$), Mel scales $\mathcal{H}$, STFT scales $\mathcal{K}$

1: **for each** minibatch $B \subset \mathcal{D}$ **do**
2:     Sample per-example messages $m \leftarrow \text{SAMPLEMSG}(B)$
    **// Stage A: Audio generation**
3:     **for each** $w_o \in B$ **do**
4:         $z \leftarrow \text{ENCODEQUANTIZE}(\hat{C}, w_o)$              $\triangleright$ stop-grad through encoder/quantizer
5:         $\hat{w} \leftarrow \hat{C}_{\text{dec}}(z; \theta)$         $\triangleright$ trainable decoder outputs watermarked audio
6:         $w \leftarrow C_{\text{dec}}(z)$         $\triangleright$ reference clean (unwatermarked) audio
7:     **end for**
    **// Stage B: Feature extractions (time/freq)**
8:     $L_{\text{time}} \leftarrow \frac{1}{|B|} \sum \|w - \hat{w}\|_1$
9:     $L_{\text{spec}} \leftarrow \text{MULTISCALEMELLOSS}(w, \hat{w}, \mathcal{H})$         $\triangleright$ Eq. (3) style
    **// Stage C: Adversarial perceptual optimization**
10:     $(L_{\text{gen}}, L_{\text{feat}}, L_{\text{disc}}) \leftarrow \text{ADVERSARIALLOSSES}(w, \hat{w}, D, \mathcal{K})$
    **// Stage D: Mask model & watermark injection/decoding**
11:     $\tilde{w} \leftarrow M(\hat{w})$         $\triangleright$ robustness & localization refinement
12:     $[\hat{y}, \hat{m}] \leftarrow D_{\text{det}}(\tilde{w}) \in [0,1]^{(2+16) \times T}$     $\triangleright$ $\hat{y}$: frame-level presence; $\hat{m}$: 16-bit payload
13:     $L_{\text{det}} \leftarrow \text{BCE}(y, \hat{y}), \quad L_{\text{payload}} \leftarrow \text{BCE}(m, \hat{m})$
14:     $L_{\text{msg}} \leftarrow \lambda_{\text{det}} L_{\text{det}} + \lambda_{\text{payload}} L_{\text{payload}}$
    **// Loss balancing (Section 2.2)**
    **// Loss balancing (EMA-rescale-clip)**
15:     $\hat{g}_{\text{dec}} \leftarrow \text{LOSSBALANCER}\Big(\{\mathcal{L}_i\}, \{\lambda_i\}; R, \rho, \beta, \tau\Big)$     $\triangleright$ $\mathcal{L}_i \in \{L_{\text{time}}, L_{\text{spec}}, L_{\text{gen}}, L_{\text{feat}}, L_{\text{msg}}\}$
16:     $\text{BACKPROPFROM}(\hat{g}_{\text{dec}} \rightarrow \theta)$
    **// Parameter updates**
17:     $\text{UPDATE}(\psi; \nabla_\psi L_{\text{disc}})$         $\triangleright$ discriminator update
18:     $\text{UPDATE}(\theta)$         $\triangleright$ decoder update
19:     $\text{UPDATE}(\phi; \nabla_\phi L_{\text{msg}})$         $\triangleright$ optimize detector for stable recovery
20: **end for**

---

GAN-based approaches, like MelGAN (Kumar et al., 2019a), enhance synthesis efficiency. These advancements established the foundation for contemporary neural audio generation techniques.

Recent studies integrate transformers and diffusion models to achieve further development for audio generation. AudioLDM (Liu et al., 2023) uses contrastive language audio pretraining (Wu et al., 2023) with latent diffusion (Rombach et al., 2022) for text-guided generation. Audio language models such as *Bark* (Suno, 2023), MusicGAN (Copet et al., 2024), and AudioLM (Borsos et al., 2023) use text-generation techniques (Radford, 2018; Brown et al., 2020a), encoding text and timbre into tokens using EnCodec (Défossez et al., 2022) and SoundStream (Zeghidour et al., 2021) for transformer-based sequence-to-sequence synthesis.

## B   Pseudocode for Train Pipeline

The following four algorithms together describe the complete training process of `GenMark`. Algorithm 1 provides the overall training pipeline, calling three key subroutines: *AdversarialLosses* (Algorithm 3), *MultiScaleMelLoss* (Algorithm 2), and *LossBalancer* (Algorithm 4). The main pipeline orchestrates data sampling, watermarked–clean audio generation, feature extraction, and adversarial optimization. Within it, *AdversarialLosses* computes generator, discriminator, and feature-matching losses; *MultiScaleMelLoss* measures spectral similarity across multiple Mel scales; and *LossBalancer* adaptively rescales and clips gradients from all losses to maintain stable training. Together, these algorithms define a unified framework for robust watermark embedding and detection.

---

**Algorithm 2** MultiScaleMelLoss

---

1: **function** MULTISCALEMELLOSS($w, \hat{w}, \mathcal{H}$)
2: $\quad L \leftarrow 0$
3: $\quad$ **for each** $h \in \mathcal{H}$ **do**
4: $\quad\quad S_w \leftarrow S_h(w), \quad S_{\hat{w}} \leftarrow S_h(\hat{w})$
5: $\quad\quad L \leftarrow L + \|S_w - S_{\hat{w}}\|_1 + \|S_w - S_{\hat{w}}\|_2^2$
6: $\quad$ **end for**
7: $\quad$ **return** $L$
8: **end function**

---

---

**Algorithm 3** AdversarialLosses

---

1: **function** ADVERSARIALLOSSES($w, \hat{w}, D, \mathcal{K}$)
2: $\quad L_{\text{disc}}, L_{\text{gen}}, L_{\text{feat}} \leftarrow 0, 0, 0$
3: $\quad$ **for each** $k \in \mathcal{K}$ **do**
4: $\quad\quad p_r \leftarrow D_k(w), \quad p_f \leftarrow D_k(\hat{w})$
5: $\quad\quad L_{\text{disc}} \mathrel{+}= \|1 - p_r\|_1 + \|p_f\|_1$ $\qquad\qquad\qquad\quad$ ▷ discriminator objective $\mathcal{L}_{\text{dic}}$
6: $\quad\quad L_{\text{gen}} \mathrel{+}= \|1 - p_f\|_1$ $\qquad\qquad\qquad\qquad\quad$ ▷ generator adversarial objective $\mathcal{L}_{\text{gen}}$
7: $\quad\quad$ **for each** layer $l \in \mathcal{S}$ **do**
8: $\quad\quad\quad f_r \leftarrow D_k^l(w), \quad f_f \leftarrow D_k^l(\hat{w})$
9: $\quad\quad\quad L_{\text{feat}} \mathrel{+}= \dfrac{\|f_r - f_f\|_1}{\mathbb{E}[f_r] + \epsilon}$ $\qquad\qquad\qquad$ ▷ feature matching $\mathcal{L}_{\text{feat}}$
10: $\quad\quad$ **end for**
11: $\quad$ **end for**
12: $\quad$ **return** $(L_{\text{gen}}, L_{\text{feat}}, L_{\text{disc}})$
13: **end function**

---

---

**Algorithm 4** LossBalancer (EMA + Rescale + Clip)

---

**Require:** losses $\{\mathcal{L}_i\}$, weights $\{\lambda_i\}$, reference $R$, EMA decay $\rho \in (0,1)$, exponent $\beta > 0$, clip threshold $\tau$, small $\varepsilon$
1: initialize $m_i \leftarrow 0, \forall i$ $\qquad\qquad\qquad\qquad\qquad\qquad\qquad\qquad$ ▷ EMA state
2: **for each** $\mathcal{L}_i$ **do**
3: $\quad g_i \leftarrow \partial \mathcal{L}_i / \partial \hat{x}$ $\qquad\qquad\qquad\qquad\qquad\qquad\quad$ ▷ compute its gradient
4: $\quad u_i \leftarrow \|g_i\|_2^\beta$ $\qquad\qquad\qquad\qquad\qquad\qquad\qquad\quad$ ▷ powered norm
5: $\quad m_i \leftarrow \rho\, m_i + (1-\rho)\, u_i$ $\qquad\qquad\qquad\qquad\quad$ ▷ track EMA of $\|g_i\|_2^\beta$
6: $\quad \tilde{g}_i \leftarrow \dfrac{R \cdot \lambda_i}{\sum_j \lambda_j} \cdot \dfrac{g_i}{m_i + \varepsilon}$ $\qquad\qquad\qquad\quad$ ▷ rescale by recent magnitude
7: **end for**
8: $g \leftarrow \sum_i \tilde{g}_i$ $\qquad\qquad\qquad\qquad\qquad\qquad\qquad\qquad\quad$ ▷ aggregate
9: $\hat{g} \leftarrow$ CLIPBYNORM$(g, \tau)$ $\qquad\qquad\qquad\qquad\qquad\qquad\quad$ ▷ clipped gradient
10: **return** $\hat{g}$

---

## C LOCALIZATION REFINEMENT MODULE

To enhance the decoder's ability to accurately localize watermarked regions and reduce false detections, we introduce two replacement strategies during training:

**Mismatched Watermark Replacement**: For each watermarked audio sample, we randomly replace 85% of its embedded watermark segments with segments carrying different (non-target) watermark messages. This helps prevent the decoder from memorizing fixed patterns and promotes generalization across diverse watermark structures.

**Random Segment Perturbation**: We divide the audio into $K$ segments and randomly select starting points for content replacement. Each selected segment (of length $2T/K$) is then altered with one of the following: 40% probability of clean (unwatermarked) waveform insertion, 20% probability

of substitution with unrelated audio, and 20% probability of silence padding. The remaining 20% is left unchanged. These manipulations simulate realistic confusion patterns that the decoder may encounter in practice.

By combining these techniques and optimizing using the decoding loss $\mathcal{L}_{\mathrm{msg}}$, the decoder is explicitly trained to focus on truly watermarked regions and reject irrelevant or misleading segments, significantly improving localization reliability during inference.

## D  ROBUSTNESS ENHANCEMENT MODULE

To improve the watermark's resilience against signal processing attacks, we introduce a robustness enhancement module composed of 11 commonly used audio transformations. These operations are applied stochastically during training, with their parameters drawn from calibrated ranges. This helps the decoder learn to preserve watermark fidelity under real-world distortions:

1. **Bandpass Filter** Removes both low- and high-frequency components while preserving a specific mid-frequency range. *Parameters:* center frequency = 2750 Hz, quality factor $Q = 0.707$

2. **Highpass Filter** Attenuates frequencies below the cutoff, simulating microphone or channel filtering. *Parameters:* cutoff frequency = 1500 Hz

3. **Lowpass Filter** Attenuates frequencies above the cutoff, emulating bandwidth-limited scenarios. *Parameters:* cutoff frequency = 500 Hz

4. **Speed Adjustment** Alters playback speed by resampling, affecting both pitch and timing. *Parameters:* speed factor $\in [0.8, 1.2]$

5. **Resampling** Converts to an intermediate sampling rate and back, introducing temporal interpolation artifacts. *Parameters:* resampled to 32kHz and then resampled back to the original frequency

6. **Boost** Multiplies the audio amplitude to simulate volume spikes or clipping. *Parameters:* boost factor = 10

7. **Duck** Reduces signal amplitude to mimic audio underpowering or suppression. *Parameters:* duck factor = 0.1

8. **Echo** Adds delayed and scaled versions of the signal to simulate reverberation. *Parameters:* delay time $\in [0.1, 0.5]$ seconds, echo volume $\in [0.1, 0.5]$

9. **Pink Noise** Adds pknk noise to simulate natural ambient environments. *Parameters:* target SNR = 20 dB

10. **White Noise** Adds flat- Gaussian noise, resembling synthetic interference. *Parameters:* target SNR = 20 dB

11. **Smoothing** Applies a moving-average filter to blur waveform details. *Parameters:* window size $\in [2, 10]$ samples

The parameter settings here are generally consistent with those used in the experiment 5.3, where additional adversarial attacks such as Encodec - using an offical 24khz encodec model, white_box - pgd attack, and black_box- square attack were also incorporated to further evaluate robustness.

In this experiment, some values were omitted from the main table due to width constraints; the missing decoding error rates (%) for the **Resample**, **Smooth**, and **MP3** columns are:

- **AudioSeal**: Resample = 4.66,  Smooth = 0.00,  MP3 = 0.00

- **WavMark**: Resample = 0.12,  Smooth = 26.33,  MP3 = 0.00

- **SilentCipher**: Resample = 7.01,  Smooth = 70.06,  MP3 = 0.00

- **Ours (`GenMark`)**: Resample = 0.15,  Smooth = 1.04,  MP3 = 0.00

# E QUALITY METRICS

NOTATION

Let $x \in \mathbb{R}^T$ denote the reference (original) audio waveform and $\hat{x} \in \mathbb{R}^T$ the corresponding generated, watermarked audio. For a dataset $\mathcal{D}$ of paired clips $\{(x_n, \hat{x}_n)\}_{n=1}^N$, all metrics are computed per clip and then aggregated over $\mathcal{D}$ (mean $\pm$ std unless stated otherwise). Unless noted, signals are resampled to a common sampling rate and time-aligned.

## E.1 DISTRIBUTIONAL METRICS

**Scale-Invariant Signal-to-Noise Ratio (SI‑SNR).** SI-SNR measures the fidelity of a target signal preserved in an estimate, while being invariant to global scaling. Let

$$\tilde{x} = x - \text{mean}(x), \quad \tilde{\hat{x}} = \hat{x} - \text{mean}(\hat{x}).$$

Project $\tilde{\hat{x}}$ onto $\tilde{x}$:

$$x_{\text{target}} = \frac{\langle \tilde{\hat{x}}, \tilde{x} \rangle}{\|\tilde{x}\|_2^2} \, \tilde{x}, \qquad e_{\text{noise}} = \tilde{\hat{x}} - x_{\text{target}}.$$

The utterance-level SI-SNR (in dB) is

$$\text{SI}-\text{SNR}(x, \hat{x}) = 10 \log_{10} \frac{\|x_{\text{target}}\|_2^2}{\|e_{\text{noise}}\|_2^2}.$$

**Range/Interpretation:** Higher is better; $+\infty$ dB only when $\hat{x}$ is a scaled copy of $x$. We report the average SI-SNR over clips.

**Kullback–Leibler Divergence (KLD).** KLD quantifies distributional mismatch between features extracted from $x$ and $\hat{x}$. Let $z = f(x)$ and $\hat{z} = f(\hat{x})$ be frame- or clip-level features (e.g., log-mel histograms or embeddings from a fixed pre-trained network). We consider two common instantiations:

*(i) Discrete/bin-wise KLD.* Let $P$ and $Q$ be normalized histograms over the same bins:

$$D_{\text{KL}}(P\|Q) = \sum_i P(i) \, \log \frac{P(i)}{Q(i)}.$$

Smoothing with a small $\varepsilon$ is used to avoid $\log 0$.

*(ii) Gaussian embedding KLD.* If we approximate clip embeddings as Gaussians with $\mathcal{N}(\mu_P, \Sigma_P)$ and $\mathcal{N}(\mu_Q, \Sigma_Q)$ in $\mathbb{R}^d$:

$$D_{\text{KL}}(\mathcal{N}_P\|\mathcal{N}_Q) = \tfrac{1}{2}\left[\text{tr}\left(\Sigma_Q^{-1}\Sigma_P\right) + (\mu_Q - \mu_P)^\top \Sigma_Q^{-1}(\mu_Q - \mu_P) - d + \ln \frac{\det \Sigma_Q}{\det \Sigma_P}\right].$$

**Range/Interpretation:** $D_{\text{KL}} \geq 0$; lower is better (0 iff the distributions match).

## E.2 PERCEPTUAL METRICS

**Fréchet Audio Distance (FAD).** FAD compares the *distribution* of embeddings computed from a large set of reference audio against that from generated audio. Let $\{\phi(x_n)\}$ and $\{\phi(\hat{x}_n)\}$ be embeddings from a fixed audio model. Estimate Gaussian statistics

$$(\mu_r, \Sigma_r) \text{ from } \{\phi(x_n)\}, \quad (\mu_g, \Sigma_g) \text{ from } \{\phi(\hat{x}_n)\}.$$

The FAD is the Fréchet distance between the two Gaussians:

$$\text{FAD} = \|\mu_r - \mu_g\|_2^2 + \text{tr}\left(\Sigma_r + \Sigma_g - 2\left(\Sigma_r \Sigma_g\right)^{1/2}\right).$$

**Range/Interpretation:** FAD $\geq 0$; lower is better and indicates a distribution of generated-audio embeddings closer to that of the reference set.

**ViSQOL (Virtual Speech Quality Objective Listener).** ViSQOL is a full-reference perceptual metric that estimates a MOS-LQO (Mean Opinion Score—Listening Quality Objective) by comparing time–frequency patches of a reference signal to those of a test signal using a spectro-temporal similarity measure. We use ViSQOL in speech mode for spoken content and audio mode for general audio. The output is mapped to a MOS-like scale.

$$\text{MOS}-\text{LQO} \in [1, 5] \quad (\text{higher is better}).$$

**Notes:** Ensuring consistent loudness normalization and resampling (e.g., $16\,\text{kHz}$ speech / $48\,\text{kHz}$ audio) improves robustness. We report the average MOS-LQO per condition.

### E.3 SEMANTIC METRIC

**CLAP (Contrastive Language–Audio Pretraining).** CLAP provides a shared embedding space for audio and language. We use it to quantify whether the *semantic content* of $\hat{x}$ matches that of $x$ (audio–audio) and/or matches a text description $t$ (audio–text).

*Audio–audio semantic similarity:*

$$s_{\text{AA}}(x, \hat{x}) = \cos\left(\frac{\phi_a(x)}{\|\phi_a(x)\|_2}, \frac{\phi_a(\hat{x})}{\|\phi_a(\hat{x})\|_2}\right) = \frac{\phi_a(x)^\top \phi_a(\hat{x})}{\|\phi_a(x)\|_2 \|\phi_a(\hat{x})\|_2}.$$

*Audio–text semantic similarity:*

$$s_{\text{AT}}(\hat{x}, t) = \cos\left(\frac{\phi_a(\hat{x})}{\|\phi_a(\hat{x})\|_2}, \frac{\phi_t(t)}{\|\phi_t(t)\|_2}\right).$$

**Range/Interpretation:** $s \in [-1, 1]$; higher is better. Optionally, retrieval-style metrics (e.g., Recall@K) can be reported using the same embeddings.

## F TPR AND FPR

First, we compared our method with the current state-of-the-art models (WavMark and AudioSeal) on several audio generation tasks, using True Positive Rate (TPR) and False Positive Rate (FPR) as evaluation metrics. TPR represents the proportion of watermarked audio correctly identified by the model, and its formula is:

$$TPR = \frac{TP}{TP + FN} \tag{8}$$

where TP refers to true positives (samples correctly identified as watermarked) and FN refers to false negatives (samples with watermarks not detected).

FPR represents the proportion of non-watermarked audio that is incorrectly classified as watermarked, and its formula is:

$$FPR = \frac{FP}{FP + TN} \tag{9}$$

where FP refers to false positives (non-watermarked samples misclassified as watermarked) and TN refers to true negatives (samples correctly identified as non-watermarked). For watermarking models, our optimization goal is to maximize TPR while minimizing FPR.

## G DETECTOR ARCHITECTURE

Inspired by the design of the AudioSeal watermark detector (San Roman et al., 2024), we implement a lightweight yet effective watermark detection model tailored for generative audio. Our detector operates directly on the raw audio waveform and outputs both a detection confidence and an optional binary message.

The architecture consists of two main components: an audio encoder and a classification head. The encoder, denoted as `self.encoder`, follows the same architectural design as the EnCodec encoder (Défossez et al., 2022), consisting of a series of downsampling convolutional blocks interleaved with residual connections. Specifically, the encoder comprises $N$ convolutional layers with progressively increasing channel dimensions and strides to reduce temporal resolution, while preserving essential information for watermark detection. To restore alignment with the input resolution, a transposed convolution layer is applied after encoding.

Following the encoder, we apply a $1 \times 1$ convolution to produce a multi-head output. The first two channels represent the confidence scores (via softmax) for the presence or absence of a watermark. The remaining $n$ channels represent the per-bit logits of the embedded binary watermark message, which are decoded via a temporal average followed by a sigmoid activation. This design allows the detector to perform both binary watermark detection and payload recovery in a unified forward pass.

## H  SUBJECTIVE EVALUATION PROTOCOL AND HUMAN STUDY INFORMATION

### H.1  MUSHRA TEST SETUP

To evaluate the perceptual audio quality of watermarked audio, we conducted a subjective study using the standardized **MUSHRA protocol** (Multiple Stimuli with Hidden Reference and Anchor), following ITU-T Recommendation BS.1534-1. This method is widely used in audio quality benchmarking and provides robust human preference data across fine-grained quality levels.

Each test session included:

- One fixed reference audio clip (original unwatermarked audio),
- Three watermarked outputs (GenMark, AudioSeal, WavMark),
- Two lossy anchors: Anchor70 (band-limited at 7 kHz), Anchor35 (band-limited at 3.5 kHz),
- One hidden reference (identical to the original, included to assess rating consistency).

Participants evaluated the samples using an interactive web-based MUSHRA interface that supports waveform visualization, looping playback, and blind randomized ordering of stimuli. The interface was customized to guide the listener through the evaluation, showing condition names only during the training phase, and hiding them during formal scoring.

We recruited **20 expert listeners** with backgrounds in audio engineering or speech synthesis. All participants voluntarily agreed to take part in the study and were informed that their responses would be used for academic research purposes only. No personally identifying information (PII) was collected. As the evaluation involved non-sensitive, low-risk listening tasks, no formal IRB approval was required.

Each participant rated **20 audio groups**, each corresponding to a different prompt. Ratings were provided on a 0–100 scale via slider interfaces, with the ability to replay any sample as needed. Anchor and reference scores were used to validate listener consistency, and all results were aggregated by condition across listeners. For quantitative analysis and comparisons, please refer to Section 5.4 of the main paper.

The testing interface was implemented as a browser-based system supporting:

- Interactive MUSHRA scoring with waveform display and audio looping,
- Randomized presentation of audio conditions per trial,
- Automated anchor generation using standard low-pass filters.

### H.2  INSTRUCTIONS PROVIDED TO PARTICIPANTS

Participants received the following instructions (translated and paraphrased from the interface):

> **Welcome to the Audio Quality Evaluation Test**
> This test assesses your subjective perception of audio quality.
> **Testing Process**:

- Left panel: Reference audio (always visible)
- Right panel: Six randomized test samples (three algorithmic outputs, two lossy anchors, one hidden reference)

**Scoring Guide**:

- 0–35: Severe degradation
- 45–60: Moderate degradation
- 61–80: Mild degradation
- 80–100: Nearly indistinguishable from reference

Please ensure a quiet environment and use high-quality headphones. Focus on high-frequency regions (e.g., fricatives like /s/, /z/) to detect perceptual differences.

## I INFORMATION-THEORETIC COMMUNICATION ANALYSIS

### I.1 CAPACITY AND SNR ANALYSIS

To establish an information-theoretic perspective on the proposed watermarking scheme, we model the embedding process as a communication channel, where the original audio acts as noise and the watermark represents the transmitted signal. Under this model, the average signal-to-noise ratio (SNR) measured across the watermarked audio samples is **–22.21 dB**, corresponding to a low-SNR regime.

According to **Shannon's capacity formula**,

$$C = W \log_2(1 + \text{SNR}), \tag{10}$$

and considering that the watermark signal is predominantly distributed below **4 kHz**, we set the channel bandwidth to

$$W = 4{,}000 \text{ Hz}.$$

This yields a theoretical channel capacity of approximately **34.76 bits per second**.

Given that the typical duration of our audio samples is about **1 second**, the achievable payload is on the order of **17.38 bits**, assuming a conservative **50% Shannon efficiency**, which is common in engineered communication systems. Based on this analysis, we select a **16-bit message length** for the watermark, which is well aligned with the theoretical capacity limits.

### I.2 SPECTRAL AND TEMPORAL DISTRIBUTION OF THE EMBEDDED WATERMARK

We further examine the spectral and temporal characteristics of the watermark embedding:

- **Spectral Domain:** More than **90%** of the energy of both clean and watermarked audio resides below **4 kHz**, and the watermark itself is constrained to the same frequency region. This alignment enhances imperceptibility and robustness while avoiding high-frequency artifacts (see Fig. 3).

- **Temporal Domain:** The watermark is **distributed across the entire audio signal**, rather than being confined to short segments. Its temporal pattern closely follows the natural structure of the host audio, thereby improving both perceptual transparency and resistance to removal (see Fig. 3).

