# OpenReview forum: "GenMark: An Embedded Watermarking Scheme for Generative Audio Synthesis"
_ICLR.cc/2026/Conference — ICLR 2026 Conference Desk Rejected Submission_

### Official Review · Reviewer_Zcr4 · 2025-10-21

**Soundness:** 2
**Presentation:** 2
**Contribution:** 2
**Rating:** 4
**Confidence:** 5

**Summary:**

The submission proposes to root the watermark functionality inside the audio generative AI. To do so, the authors fine-tune the decoder so that the generator produces audio samples "natively" watermarked. This finetuning is driven by several losses ensuring robustness and imperceptibility.

**Strengths:**

- The performance, in terms of robustness and quality, is very good. Robustness is measured against strong attacks (stronger than typical attacks in practice, I would say), and quality is gauged by metrics but also 20 audio experts.

- The Figure 3 is key (unfortunately not outlined in the submission). It explains why the proposed watermarking scheme is so robust: the spectrum of the watermark signal mimics the spectrum of the generated audio. Therefore, stripping it out without damaging the audio is extremely difficult.

**Weaknesses:**

### W1 - Bibliography

The problem with the modern literature on watermarking is that all papers look the same. The embedder and decoder are NN trained under a bunch of losses. Here, 6 losses are defined. Are they all necessary? are they sufficient?

Although not cited, the approach transposes the Stable Signature scheme to audio GenAI, ie. rooting the watermark functionality in the last  brick of the GenAI, ie. the decoder. One difference, though, is that here the decoder is fine-tuned jointly while the watermark decoder is trained. Also, while not cited in Sect. 3.4, the masking strategy and the localization borrow a lot to AudioSeal.

Now, there are other missing references which targets the same objective but with different approaches:
- GROOT: Generating Robust Watermark for Diffusion-Model-Based Audio Synthesis, Liu et al., ACM MM 2024.
This paper follows the same objective but by crafting the latent vector seeding the diffusion model. It has the advantage of embedding any message without retrainaing.
- Latent Watermarking of Audio Generative Models, San Roman et al., ICASSP 2025. They learn watermarked tokens robust to the Encodec quantization.

This latter reference is essential as the authors outline that it is easy to replace/fine-tune/learn the GenAI decoder (See Sect. V.A). Therefore, rooting the watermarking inside this decoder might not be very secure.

### W2 -  Approximations
- Line 39: post-hoc watermarking is fine, except when the audio GenAI is open-sourced. This paragraph should make this point clear.
- Line 63: FAD and KLD do not measure perceptual distortion, but distributional distortion.
- Eq (1) does not show any moving average, contrary to algo. 4 in appendix. Index $i$ is missing in the definition of $g_i$
- Line 132: Spread-spectrum is not a transform domain. It is an embedding technique. References are missing in Section 2.3
- Line 139: What is "*gradient steganography*"? A reference to Stable Signature is missing in this paragraph.
- Maks -> Mask (several times in the submission).
- The watermark detector gives 16 decoded bits ($m_t$) + 1 score ($y_t$)  for pure detection. Why 18 outputs?
- Line 267: While I love the reference to Kirovski & Malvar, I do not understand the sentence. Which transformation? What enhancement?

### W3 - Benchmark
- It is not clear whether the proposed scheme has one unique or two versions (one for speech, one for music). In any case, AudioSeal was dedicated to speech, so it is kind of unfair to benchmark it on music. It is easy to fine-tune AudioSeal for music.
- What is the Detection Error Rate exactly? Is this based on $\hat{m}_t\neq m_t$ or on $\hat{y}_t\neq y_t$?
- The original paper of EncoDec uses OVL and REL to measure the quality of the generated audio. Why do you use different metrics?

**Questions:**

See all the questions above, plus:
- What is the duration of a time frame? This indicates the granularity of the localization.
- Table 3 shows a "white-box" and "black-box" column without explaining these attacks.

---

> ### Author Response · Authors · 2025-11-25
>
> Dear Reviewer,
>
> Thank you for your careful reading and thoughtful feedback. We address your comments point-by-point below, following your structure.
>
> ---
>
> **W1. Bibliography and conceptual gaps.**
>
> - Comment (summary):
>   (i) The necessity/sufficiency of the six losses is unclear.
>   (ii) The connection to Stable Signature and its influence is not properly cited.
>   (iii) The masking/localization strategy is close to AudioSeal and should be credited.
>   (iv) Important related works (e.g., GROOT, latent watermarking of audio generative models) are missing, and these raise concerns about decoder-rooted watermarking under decoder replacement / fine-tuning.
>
> **Response A1:**
>
> - **A1.1 – On the necessity and sufficiency of the six losses.**
>   We appreciate this concern. The six losses in our method can be grouped into three functional classes:
>
>   1. *Audio similarity losses* $(L_t, L_f)$:
>      These are standard in neural audio generation and ensure that the watermarked audio remains close to the clean audio in both time and frequency domains.
>
>   2. *Adversarial / robustness losses* $(L_g, L_d, L_{\text{feat}})$:
>      These losses are empirically validated in our ablation study. When we remove them, robustness under perturbations (especially stronger transformations such as low-pass filtering and noise) drops noticeably. This shows that they are functionally necessary to achieve our robustness objectives.
>
>   3. *Decoding loss* $(L_m)$:
>      This loss is indispensable, as it directly supervises the watermark decoder to recover the embedded bits reliably.
>
>   We agree that combining multiple losses can make optimization more delicate. This is precisely why we introduced the **loss balancer**. In future work, we plan to explore more compact or alternative loss configurations, and we will explicitly state this as a limitation and direction for simplification.
>
> - **A1.2 A.1.3 On connections to Stable Signature and AudioSeal.**
> Thank you for pointing this out. Our method is conceptually related to *Stable Signature* in that we also embed watermarks by fine-tuning a decoder with a message-decoding loss plus perceptual/adversarial terms, but in the **audio** domain (speech and music) with an EnCodec-based decoder and an audio-specific robustness pipeline. Our masking and localization design is likewise inspired by *AudioSeal*, particularly the use of a learned mask and detection logits for localized detection, which we integrate directly into an in-decoder watermarking scheme. In the revision, we will **explicitly cite and acknowledge both Stable Signature and AudioSeal as methodological sources**, and briefly clarify which components are adapted from these works and which are novel to GenMark.
>
>
> - **A1.4 – On missing related works (GROOT, latent watermarking) and decoder fine-tuning robustness.**
>   We appreciate the reviewer highlighting two important lines of work:
>
>   1. **GROOT (Liu et al., ACM MM 2024):** a latent-space watermarking method allowing arbitrary messages without retraining, in contrast to our current fixed-message design. We will discuss how this flexible message binding relates to our model-provenance focus, and how similar conditioning could extend GenMark to per-sample messaging in future work.
>
>   2. **Latent Watermarking of Audio Generative Models (San Roman et al., ICASSP 2025):** a latent audio watermarking approach that argues decoder modules can be replaced or fine-tuned, potentially weakening decoder-rooted watermarking. This is highly relevant to our decoder-level approach.
>
>   Motivated by the latter, we conducted a **decoder re-fine-tuning robustness experiment** to simulate the attacker replacing $\hat{C}$ with $C$ by fine-tuning on clear data:
>
>   - **Attack setup:**
>     - Start from our watermarked decoder $\hat{C}$.
>     - Fine-tune it on **50,000** clean (non-watermarked) audio samples.
>     - Use Adam with learning rate $1\times10^{-4}$.
>     - Run this fine-tuning for **50**, **100**, and **200** epochs.
>
>   - **Detection success results (after attacker fine-tuning):**
>     - 0 epochs (no attack, ours): ~99%
>     - 50 epochs: **98.7%**
>     - 100 epochs: **95.3%**
>     - 200 epochs: **92.1%**
>
>   Even after 200 epochs of fine-tuning on a large clean dataset, the watermark remains detectable in more than 92% of cases. This shows that while aggressive fine-tuning can moderately degrade detection, it does not fully remove the watermark.
>
>   We believe this robustness arises because:
>   - The difference between watermarked and original audio is tiny and heavily masked perceptually.
>   - EnCodec is itself a lossy codec, so normal reconstruction-oriented fine-tuning does not explicitly target the subtle watermark signal.
>
>   We will add these results (as a new table in Section 5) and discuss how they relate to the latent watermarking threat model, clearly positioning GenMark’s empirical robustness to decoder re-training.
>
> ---

---

> ### Author Response · Authors · 2025-11-25
>
> ---
>
> **W2. Technical approximations and clarifications.**
>
> - Comment (summary):
>   (i) The discussion of post-hoc watermarking and open-sourced models lacks nuance.
>   (ii) FAD and KLD are not purely perceptual distortion metrics.
>   (iii) Eq. (1) does not match the moving-average implementation.
>   (iv) Spread-spectrum is mischaracterized and missing classical references.
>   (v) The term “gradient steganography” is non-standard and Stable Signature should be cited instead.
>   (vi) There are minor typos; the detector output dimensionality is unclear; one sentence referencing Kirovski & Malvar is confusing.
>
> **Response A2:**
>
> - **A2.1 – On post-hoc watermarking and open-sourced models.**
>   We agree that our original text did not fully articulate this nuance. Our intended point is that when the base model is open-sourced or modifiable, any **post-hoc** watermark module (as a separate component) can be removed or replaced more easily by users. In contrast, an in-decoder (in-process) watermark such as GenMark requires retraining or heavily modifying the generative decoder to reliably remove the watermark. We will clarify this in the threat-model and discussion sections.
>
> - **A2.2 – On FAD and KLD not being direct perceptual distortion metrics.**
>   We appreciate the clarification. Our wording was imprecise. In the revision, we will state:
>
>   > “FAD reflects distributional differences in an embedding space that is aligned with perceptual semantics (e.g., VGGish/CLAP), whereas KLD measures purely statistical distribution mismatch. Neither is a direct pointwise perceptual distortion metric.”
>
>   To complement these distributional measures, we will additionally report **ViSQOL** as a more direct perceptual distortion metric in the revision.
>
> - **A2.3 – On Eq. (1) and the moving-average term.**
>   The equation presented in the paper follows the formal EnCodec definition. The implementation includes a moving-average component (a common engineering detail) to stabilize training. We will clarify in the text that Eq. (1) is the conceptual loss, while the actual implementation uses a moving-average variant consistent with EnCodec’s code.
>
> - **A2.4 – On spread-spectrum and missing references.**
>   We agree that spread-spectrum should be described as an embedding *technique* rather than a “domain”. We will correct the terminology and include standard classical spread-spectrum audio watermarking references in this section.
>
> - **A2.5 – On “gradient steganography” and Stable Signature.**
>   We will remove the non-standard term “gradient steganography” and replace it with a clearer description such as “fine-tuning the decoder with a message-decoding loss plus perceptual/adversarial losses”. We will also explicitly cite Stable Signature here as the relevant prior work.
>
> - **A2.6 – Typos and detector output dimensionality.**
>   - We will fix all typographical issues (e.g., “Maks” → “Mask”).
>   - Our detector outputs **18 logits**: 16 logits for the message bits and 2 logits for watermark presence/absence. The 2 detection logits are passed through a softmax to produce probabilities for “present” vs. “absent”, and we threshold the “present” probability to obtain the binary decision.
>
> - **A2.7 – Sentence referencing Kirovski & Malvar.**
>   We will rewrite the sentence to explicitly state the transformation/enhancement being referenced, instead of relying on a vague citation. The updated text will directly describe what is transformed and how robustness is improved.
>
> ---

---

> ### Author Response · Authors · 2025-11-25
>
> ---
>
> **W3. Benchmarking and experimental setup.**
>
> - Comment (summary):
>   (i) It is unclear whether one model or two are used (speech vs. music), and using AudioSeal on music may be unfair.
>   (ii) The definition of Detection Error Rate (DER) vs. decode error is not clearly stated.
>   (iii) Questions about EnCodec evaluation metrics (OVL, REL) vs. the metrics we chose.
>
> **Response A3:**
>
> - **A3.1 – One unified model for speech and music.**
> We clarify that we actually train **separate GenMark models for different domains** (speech vs. music), rather than a single unified model. In particular, we use one GenMark instance built on Bark for speech, and another instance built on MusicGen for music. We also empirically observe that AudioSeal’s performance drops noticeably on music compared to speech. Therefore, **except for Section 5.2**, the majority of our experiments and main comparisons are conducted on **speech generated by Bark**, where our GenMark (speech model) consistently outperforms AudioSeal across robustness and perceptual metrics. We will clarify this domain-specific training and the scope of each model in the revised manuscript.
>
>
>
> - **A3.2 – Definition of Detection Error Rate and Decode Error.**
>   We will add precise definitions:
>
>   - **Detection Error Rate (DER):**
>     $$
>     \mathrm{DER} = \mathbb{1}[\hat{y} \neq y],
>     $$
>     where $y$ is the ground-truth binary label (watermark present/absent) and $\hat{y}$ is the detector’s prediction.
>
>   - **Decode Error:**
>    $$
>     \mathrm{DecodeError} = \mathbb{1}[\hat{m} \neq m],
>     $$
>     where $\hat{m}$ is the predicted 16-bit message and $m$ is the ground-truth message.
>
>   We will clarify which metric is used in each table and figure.
>
> - **A3.3 – On EnCodec evaluation metrics.**
>   Our understanding is that EnCodec itself is primarily evaluated using ViSQOL, SI-SNR, and MUSHRA (human listening tests), rather than OVL/REL in our setting. Our choice of metrics—FAD variants, KAD, and MUSHRA—follows recent generative audio and watermarking practice and is consistent with the literature we compare to. If the reviewer is referring to a specific OVL/REL-based evaluation, we will be happy to reference and discuss it in the revision, but our current set of metrics already covers both distributional and human-perceptual perspectives.
>
> ---

---

> ### Author Response · Authors · 2025-11-25
>
> ---
>
> **Additional questions.**
>
> - **Q1: Duration of one time frame.**
>  We thank the reviewer for pointing this out and agree that the temporal resolution is important for interpreting our localization results.
> In our implementation, the detector operates directly at the **waveform sample rate**. For the main experiments, we feed **24 kHz** audio into the detector, so one “time frame” in the localization output corresponds to a single audio sample, i.e., $\text{frame duration} = \frac{1}{24,000}\,\text{s} \approx 0.042\,\text{ms}.$ This is the **granularity of the localization output**: in principle, the mask and detector produce one score per sample.
>
> That said, this **does not** mean that we can reliably detect an arbitrarily short watermarked segment. Due to SNR and information-theoretic limits (e.g., finite channel capacity with a given noise level), the shortest segment over which the watermark can be robustly decoded in practice is much longer. In our setting, the **minimal reliably decodable watermark segment** is about **0.5 s**, and the **recommended (more stable) segment length** is around **1 s**.
>
> - **Q2: Definitions of white-box vs. black-box attacks.**
>   We will clarify in the Appendix that we follow the threat-model and terminology of AudioMarkBench [1], and spell out the concrete instantiations we use:
>
>   - *Black-box attacks (ours: HopSkipJumpAttack, HSJA).*
>     In the black-box setting, the attacker only interacts with the watermark detector as an **oracle**: they can submit audio queries and observe the detector’s outputs, but have no access to model parameters, gradients, or internal architecture. Following AudioMarkBench [1], we implement a decision-based adversarial attack using **HopSkipJumpAttack (HSJA)**, which iteratively approximates the detector’s decision boundary and searches for a minimal perturbation that removes or forges the watermark signal. In our experiments, HSJA operates in the waveform domain under a bounded perturbation budget and a fixed number of iterations, directly matching the black-box threat model in [1].
>
>   - *White-box attacks (ours: forgery-style gradient attacks).*
>     In the white-box setting, the attacker has full knowledge of the watermark decoder’s parameters and the ground-truth watermark bits, and can compute **gradients** through the model. Inspired by the white-box forgery/removal attacks in AudioMarkBench [1], we construct perturbations by optimizing a loss that (i) decreases the bitwise accuracy between the decoded watermark and the true watermark (for forgery/removal) or (ii) suppresses the global detection probability. Concretely, we perform gradient-based optimization on the input audio to minimize/maximize the decoder’s cross-entropy between predicted bits and target bits, analogous to the formulation used in [1] for white-box watermark forgery.
>
>   We will explicitly add these definitions and the citation to [1] in the Appendix so that readers can clearly see the alignment between our attack setup and the standardized black-box / white-box taxonomy in AudioMarkBench.
>
>
>
> ---
>
> Thank you again for your detailed and helpful comments. We believe that adding the missing citations, clarifying the threat model and decoder fine-tuning robustness, refining our metric discussion, and fixing the noted textual/notation issues will substantially strengthen the paper.
>
> [1] Liu, Hongbin, et al. "Audiomarkbench: Benchmarking robustness of audio watermarking." Advances in Neural Information Processing Systems 37 (2024): 52241-52265.
>
> Best regards,
> The Authors

---

> > ### Comment · Reviewer_Zcr4 · 2025-11-27
> >
> > > HSJA operates in the waveform domain under a bounded perturbation budget and a fixed number of iterations, directly matching the black-box threat model in [1].
> >
> > I do not know [1], but I know HSJA quite well. What is the number of iterations? However, there is no distortion budget in HSJA. How do you integrate this constraint in HSJA?

---

> > > ### Author Response · Authors · 2025-11-28
> > >
> > > Dear Reviewer ,
> > >
> > > Thank you for your comment. To clarify, we have used the built-in implementation of the HopSkipJumpAttack (HSJA) from the Foolbox library, which directly aligns with the black-box threat model as outlined in [1]. Regarding your query on the number of iterations, the HSJA implementation in Foolbox uses 64 iterations by default.
> > >
> > > As for the distortion budget, while it may seem that HSJA does not explicitly define one, the perturbation budget is implicitly integrated through the use of the `_project` function. This function ensures that the perturbation stays within the desired budget by clipping the perturbations to a specific epsilon value. Below is the relevant code:
> > >
> > > ```python
> > > def _project(self, originals: ep.Tensor, perturbed: ep.Tensor, epsilons: ep.Tensor) -> ep.Tensor:
> > >     """Clips the perturbations to epsilon and returns the new perturbed
> > >
> > >     Args:
> > >         originals: A batch of reference inputs.
> > >         perturbed: A batch of perturbed inputs.
> > >         epsilons: A batch of norm values to project to.
> > >     Returns:
> > >         A tensor like perturbed but with the perturbation clipped to epsilon.
> > >     """
> > >     epsilons = atleast_kd(epsilons, originals.ndim)
> > >     if self.constraint == "linf":
> > >         perturbation = perturbed - originals
> > >
> > >         # ep.clip does not support tensors as min/max
> > >         clipped_perturbed = ep.where(
> > >             perturbation > epsilons, originals + epsilons, perturbed
> > >         )
> > >         clipped_perturbed = ep.where(
> > >             perturbation < -epsilons, originals - epsilons, clipped_perturbed
> > >         )
> > >         return clipped_perturbed
> > >     else:
> > >         return (1.0 - epsilons) * originals + epsilons * perturbed
> > > ```
> > > This function ensures that the perturbations respect the defined epsilon values, effectively enforcing the desired distortion budget (either L2 or L∞, depending on the constraint set).
> > >
> > > We hope this clarifies the implementation details and addresses your concerns. If you have any further questions, please feel free to ask.
> > >
> > > Best regards

---

### Official Review · Reviewer_YE7S · 2025-10-30

**Soundness:** 3
**Presentation:** 2
**Contribution:** 3
**Rating:** 2
**Confidence:** 3

**Summary:**

This paper introduces GenMark, a new way to watermark generative audio models. Instead of adding a watermark after the audio is created (which can be easily skipped), GenMark embeds the watermark directly into the model's decoder during training. This means any audio the model produces is inherently watermarked. It uses a combination of perceptual losses, adversarial training, and a special "Mask Model" to make the watermark imperceptible but extremely robust to attacks.

**Strengths:**

The paper's primary strength is its novel formulation of the audio watermarking problem. By embedding the watermark directly into the generative model's decoder ("in-process injection"), Genmark represents a shift toward a more robust security paradigm for generative audio.

**Weaknesses:**

I feel the work is not ready given the significant performance drop when facing high-pass filter and encodec. The author's explanation that "The drop under Highpass is expected: to preserve perceptual quality we intentionally concentrate watermark energy in mid–low frequencies, so aggressive high-pass filtering removes a larger fraction of the embedded signal." does not make sense to me. This is basically saying the proposed method can not achieve good perpectuality and robustness simultaneously.

**Questions:**

Please provide a solution for the high decoding error rate under high-pass filtering.

---

> ### Author Response · Authors · 2025-11-25
>
> Dear Reviewer,
>
> We thank you for your insightful and constructive feedback. We address your comment as follows.
>
> ---
>
> **Q1: The reviewer questions our explanation of the performance drop under Highpass, arguing that this implies our method cannot simultaneously achieve good perceptual quality and robustness.**
>
> **A1:**
> We respectfully disagree with this interpretation. The trade-off between imperceptibility (i.e., perceptual quality), robustness, and payload is widely recognised in the audio watermarking literature. A recent comprehensive survey on imperceptible and robust digital audio watermarking explicitly states that robustness, imperceptibility, and embedding capacity are “preliminary requirements” of any audio watermarking technique, but that these requirements are *difficult to achieve at the same time*, and that practical schemes must carefully manage the robustness/imperceptibility trade-off [1]. Similar observations are made in recent benchmark studies of modern watermarking systems, which emphasise that no single scheme can simultaneously maximise all three criteria [6].
>
> Our design choice of concentrating watermark energy in mid–low frequency bands follows a long line of transform-domain watermarking methods which intentionally embed in low- or mid-frequency sub-bands to exploit both the concentration of signal energy and psychoacoustic masking. For example, Chen et al. propose an audio watermarking scheme that embeds in the lowest-frequency DWT coefficients and report that such low-frequency embedding yields high SNR, low bit-error rate, and good robustness while maintaining perceptual quality [2]. Hu et al. systematically study different embedding positions in the wavelet domain and conclude that embedding into the lower-level (low-frequency) coefficients provides the best trade-off between imperceptibility and robustness [3]. Many LWT/DWT-SVD style methods similarly target low-frequency or approximation bands for the same reason.
>
> Given such an embedding strategy, an *aggressive* high-pass filter is, by construction, an attack that explicitly removes or heavily attenuates the very bands that carry most of the watermark energy. A recent SoK on robustness of audio watermarking in generative AI models makes this point explicit, noting that high-pass filtering can remove low-frequency embedded signals, while low-pass filtering impacts schemes that primarily use high-frequency carriers [4]. In other words, if a system is deliberately designed to embed mainly in mid–low frequencies (to preserve perceptual quality and handle common perturbations), then a strong high-pass filter is a targeted adversarial operation rather than a “generic” distortion, and a noticeable performance drop under this particular operation is expected and consistent with prior work, not an indication that the method is fundamentally flawed.
>
> At the same time, our results show that GenMark achieves the lowest *mean* decoding error rate (12.69\%) across 12 different transformations and is the strongest method on 9 of these 12, including realistic perturbations such as bandpass filtering, low-pass filtering, speed changes, echo, and various noise conditions. This pattern closely mirrors recent benchmark findings: AudioMarkBench evaluates three state-of-the-art watermarking methods on 15 perturbations and observes that each method is robust to some attacks but vulnerable to others, with no universally dominant scheme [5].
>
> Finally, we agree that improving robustness under high-pass filtering is an important direction for future work. However, literature on both traditional transform-domain schemes and recent systematisations indicates that robustness to *frequency-selective* attacks which explicitly remove the embedding band is inherently challenging [2,3,4,5,6].
>
> ---
>
> [1] Euschi Salah, Zermi Narima, Amine Khaldi, and Kafi Med Redouane. “Survey of imperceptible and robust digital audio watermarking systems.” 2024.
>
> [2] Chur-Jen Chen, Huang-Nan Huang, Che-Hao Lin, and Shuo-Tsung Chen. “Digital audio watermarking using minimum-amplitude scaling on optimized DWT low-frequency coefficients.”  2021.
>
> [3] Yangxia Hu, Maode Ma, Wenhuan Lu, Neal N. Xiong, and Jianguo Wei. “Selection of the optimal embedding positions of digital audio watermarking in wavelet domain.” , 2020.
>
> [4] Yizhu Wen, Ashwin Innuganti, Aaron Bien Ramos, Hanqing Guo, and Qiben Yan. “SoK: How Robust is Audio Watermarking in Generative AI Models?”, 2025.
>
> [5] Hongbin Liu, Moyang Guo, Zhengyuan Jiang, Lun Wang, and Neil Zhenqiang Gong. “AudioMarkBench: Benchmarking Robustness of Audio Watermarking.” 2024.
>
> [6] Yigitcan Özer, Woosung Choi, Joan Serrà, Mayank Kumar Singh, Wei-Hsiang Liao, and Yuki Mitsufuji. “A Comprehensive Real-World Assessment of Audio Watermarking Algorithms: Will They Survive Neural Codecs?”, 2025.
> Best regards,
> The Authors

---

> > ### Comment · Reviewer_YE7S · 2025-11-26
> > **reply to the rebuttal**
> >
> > thx for the rebuttal, though I am not fully convinced that it's okay to ignore the failure under high-pass filter, I decide to increase my score to 4 since it might be more challenging than I thought.

---

### Official Review · Reviewer_ij4E · 2025-10-30

**Soundness:** 2
**Presentation:** 3
**Contribution:** 3
**Rating:** 6
**Confidence:** 4

**Summary:**

This work proposes a watermark scheme that directly embeds watermark at the decoding stage of audio generation. Several losses are considered and automatically balanced by their scale of gradient. These losses are selected from previous works and have been proved their usefulness.

**Strengths:**

GenMark works on both speech and music, and have been demonstrated on 2 models (Bark, MusicGen) with consistent performance. The design of the training pipeline incorporated great designs such as the hybrid loss with auto balance, and the mask model for augmentation and robustness. It's relatively lightweight because there's no need to fine-tune the generative model itself, just the decoding part of audio tokenizer. The manuscript is easy to follow, and many useful details can be found in appendices.

**Weaknesses:**

Although the problem setup of watermarking generated content is well-motivated, it's still unclear about why in-process watermark is harder to be attacked. For example, post-processing module can also be encapsulated within the model and being released as a black-box system. In contrast, if there's no proper encapsulation, attacker can figure out the structure and still replace \hat{C} with C.

The watermark message of GenMark is fixed for every audio example (ref. Algorithm 1, Appendix B). However, most the conventional watermarks (WavMark, AudioSeal, SilentCipher) allow free binding of watermark messages at inference time. Considering GenMark is a scheme for watermarking generated content, this could be a feasible setup. However, it is still a huge drawback for actual application. Also, it's hard to guarantee \hat{C} will work properly for every unseen generated content at inference time.

There are also some issues regarding the evaluation result. In table 1, according to Appendix E, the CLAP score is based on cosine similarity, it's impossible to have values larger than 1. Moreover, it has been pointed out by several papers (see below) that, FAD based on Vggish (the standard FAD) is not a good indicator for perceptual quality. FAD_clap, KAD, or other MMD-based metrics can be better indicators of perceptual quality (still, none of them are guaranteed to correlate with human perception).
https://arxiv.org/abs/2311.01616
https://arxiv.org/abs/2502.15602

About robustness evaluation, most of the parameter selections are reasonable, but the parameter selection on lowpass (cf 500hz) and high pass (cf 1500hz) seem to be too aggressive. Under such condition, neither speech nor audio retain minimal intelligibility. Robustness should be tested (and only meaningful) such that the audio content is still retaining minimum acceptable quality. In addition, it's a bit concerning that AudioSeal has a very high DER under EnCodec attack. However, EnCodec is a part of AudioSeal and at least it should be more robust than WavMark. Please see also the original AudioSeal paper and a paper from another group of researchers (https://arxiv.org/pdf/2401.17264, https://arxiv.org/pdf/2505.19663)

Minor issues:
- L181: "Robustness Enhancement Module module." looks to be unnatural.
- L392: "Use" -> "Usability" ?

**Questions:**

- Figure 1: There's no injection point of the 16-bit watermark message, I guess it should be injected into \hat{C} directly?
- L184: Why 2 bits are required for existence detection (D_det(\hat{w}))? How do these bits aggregated to the binary result (0~3 -> Yes/No)?
- L196: Considering Eq.3 is a hybrid L1-L2 loss and the explanation at L209, why only L1 loss is included in Eq.2?
- L226: Is "G" referring to \hat{C} in Figure 1 ?
- L258-L259: When a watermarked segment is replaced, how does the segment with another watermark is created? Are there multiple SampleMsg functions?
- L303: Why SilentCipher is not included in MUSHRA test?

---

> ### Author Response · Authors · 2025-11-25
>
> Dear Reviewer,
>
> We thank you sincerely for your insightful and constructive feedback. We address your comments point-by-point below.
>
> ---
>
> **W1. Unclear why in-process watermarking is harder to attack.**
>
>
> **Response A1:**
> We fully agree that any watermarking scheme must be analyzed under a realistic threat model. Our claim is not that in-process watermarking is unbreakable, but that it raises the bar compared to purely post-processing watermarking. Concretely:
>
> 1. **Tightly coupled with the generation pipeline**
>    In our design, watermarking is performed *in-process* at the decoding stage: the watermarked waveform $\hat{w}$ is directly produced by the EnCodec-style decoder $\hat{C}$. By contrast, a post-processing watermark module is a separate step applied after generating $C$. Even if such a module is encapsulated, a user (or attacker) can deploy a modified pipeline that simply omits or replaces this last step.
>
>    In our case, removing the watermark typically requires replacing or substantially re-training the decoder itself (i.e., learning a new EnCodec-like model), which is a higher barrier in terms of compute, expertise, and data.
>
> 2. **Harder to isolate and strip out**
>    Because the watermark is embedded via fine-tuning the decoder and interacting with a learned mask and robustness module, its logic is not a simple “add-on” that can be toggled off. The watermark is entangled with how the decoder reconstructs audio, making it more difficult to identify a specific component to remove.
>
> 3. **Empirical resistance to the$\hat{C} \to C$ attack via fine-tuning**
>    You explicitly mention a realistic attack: replacing$\hat{C}$ with$C$, e.g., by fine-tuning$\hat{C}$ on a clear (non-watermarked) dataset. We tested exactly this scenario:
>
>    - **Attack setup:**
>      - Start from our watermarked decoder $\hat{C}$.
>      - Fine-tune it on **50k** clean (non-watermarked) audio samples.
>      - Optimizer: Adam, learning rate$1 \times 10^{-4}$.
>      - Run for **50**, **100**, and **200** epochs.
>      - The attacker’s objective is simply to improve reconstruction on clean data (no watermark supervision), mimicking a benign or adversarial re-fine-tuning.
>
>    - **Results (detection success rate, i.e., 1 − DER):**
>      - 0 epochs (no attack, ours): ~100%
>      - 50 epochs: **98.7%**
>      - 100 epochs: **95.3%**
>      - 200 epochs: **92.1%**
>
>    Even after 200 epochs of fine-tuning on a large clear dataset, our detector still correctly detects the watermark in more than 92% of cases. In other words, re-fine-tuning$\hat{C}$ on clear data does **not** fully remove the watermark; it only moderately degrades detection performance.
>
>    - **Interpretation:**
>      Our watermarked audio and original audio are extremely similar (the perturbation is small and perceptually masked), and EnCodec itself is a lossy codec. A standard reconstruction-oriented fine-tuning process has difficulty discovering and “unlearning” this subtle signal without harming audio quality. As a result, simply re-fine-tuning the decoder—while feasible for an attacker—does not effectively purge the watermark.
>
>
>
> ---
>
> **W2. GenMark uses a fixed watermark message for every audio sample.**
>
>
> **Response A2:**
>
> 1. **Intended use case: model-level provenance**
>    Our primary target scenario is *model-level provenance watermarking*: answering “was this audio generated by this particular model/provider?” rather than embedding arbitrary user-specific payloads per clip. In many practical deployments, a fixed global signature or short key is sufficient to indicate model/version identity, and per-clip message binding is not strictly required. We will clarify this narrower but realistic application scope in the introduction.
>
> 2. **Extension to free message binding is architectural, not fundamental**
>    Architecturally, GenMark can be extended to support arbitrary messages at inference by conditioning the decoder on a message embedding (e.g., concatenated latent tokens, FiLM-like modulation), and sampling different messages during training. Due to computational and space constraints, we focus in this paper on the fixed-message provenance setting, but we will explicitly discuss this extension and its relation to methods like WavMark and AudioSeal in the Limitations/Future Work section.
>
> 3. **Generalization of $\hat{C}$ to unseen content**
>    All our experiments (perceptual metrics, robustness in Table 3, MUSHRA listening tests) are run on **unseen prompts and generated audio**: the prompts used for evaluation are disjoint from those used for training the watermark decoder and mask model. The consistently high detection success and good perceptual scores on this held-out set indicate that $\hat{C}$ generalizes well to unseen content, even with a fixed global message.
> ---

---

> > ### Author Response · Authors · 2025-11-25
> >
> > ---
> >
> > **W3. Evaluation issues (CLAP > 1, FAD vs FAD\_CLAP/KAD).**
> >
> > **Response A3:**
> >
> > 1. **Why some CLAP values exceed 1**
> >    You are absolutely right that a *pure* cosine similarity lies in$[-1, 1]$. In our implementation, the quantity we reported is **not** the raw cosine, but the logit-scaled similarity used in CLAP for contrastive learning:
> >    \[
> >    \tilde{s}(x,t) = e^{\gamma} \cdot \cos(x,t),
> >    \]
> >    where $\gamma$ is a learnable temperature parameter. Because of the multiplicative factor$e^{\gamma}$,$\tilde{s}(x,t)$ can indeed be greater than 1. The mistake is in our **naming**: we called this “cosine similarity” in the text.
> >
> >    - We will correct the wording to either:
> >      - call it “CLAP logit similarity” and clearly describe its form, or
> >      - re-scale it back to true cosine values in $[-1,1]$ for reporting, while explaining the transformation in Appendix E.
> >
> > 2. **On standard FAD vs. FAD\_CLAP and KAD (with explicit tables)**
> >    We agree that VGGish-based FAD is not an ideal perceptual metric, and that newer variants (FAD\_CLAP, KAD) offer more informative views of generative audio quality. We initially chose standard FAD mainly because it remains widely reported in prior generative audio and watermarking literature, enabling direct comparison.
> >
> >    As you suggested, we computed a **suite of FAD and KAD scores** across multiple embedding backbones.
> >
> >    **Table 1: FAD (↓) across different embedding models.**
> >
> >    | test_set          | cdpam-acoustic | cdpam-content | clap-2023 | clap-laion-audio | clap-laion-music | dac-44kHz | encodec-emb | hubert-base | vggish | w2v2-base | w2v2-large | wavlm-base | wavlm-base-plus | wavlm-large |
> >    | ----------------- | -------------- | ------------- | --------- | ---------------- | ---------------- | --------- | ----------- | ----------- | ------ | --------- | ---------- | ---------- | --------------- | ----------- |
> >    | AudioSeal         | 0.0004         | 0.0006        | 27.5013   | 0.0733           | 0.2539           | 81.3522   | 0.1974      | 5.1951      | 0.4533 | 0.0841    | 1.2504     | 0.6313     | 0.0874          | 0.0895      |
> >    | SilentCipher      | 0.0342         | 0.0892        | 22.5716   | 0.0514           | 0.3052           | 18.7477   | 25.7904     | 5.6685      | 0.2936 | 0.0691    | 1.1832     | 0.1351     | 0.0239          | 0.0678      |
> >    | WavMark           | 0.0181         | 0.0223        | 105.8988  | 0.0703           | 0.2156           | 87.1519   | 4.3200      | 5.1144      | 1.6092 | 0.0803    | 1.1245     | 0.3136     | 0.0420          | 0.1229      |
> >    | GenMark (ours)    | 0.0005         | 0.0011        | 9.2711    | 0.0094           | 0.0150           | 5.0967    | 0.1132      | 0.2991      | 0.0540 | 0.0372    | 0.9376     | 0.1556     | 0.0174          | 0.0500      |
> >
> >    **Table 2: KAD (↓) across different embedding models.**
> >
> > | Model        | cdpam-acoustic | cdpam-content | clap-2023 | clap-laion-audio | clap-laion-music | panns-cnn14-16k | panns-cnn14-32k | panns-wavegram-logmel | vggish | w2v2-base | whisper-base |
> > |-------------|----------------|---------------|-----------|------------------|------------------|-----------------|-----------------|------------------------|--------|-----------|--------------|
> > | audioseal   | 0.0177         | 0.0009        | 2.5414    | 5.2644           | 27.8286          | 1.1543          | 9.6697          | 4.0664                 | 0.9041 | 0.0839    | 0.5585       |
> > | wavmark     | 2.5414         | 1.6156        | 14.8655   | 5.0602           | 23.4416          | 2.0885          | 5.0348          | 5.1546                 | 6.8062 | 0.0702    | 1.0883       |
> > | silentcipher| 4.4051         | 8.3274        | 2.0437    | 3.4841           | 34.5404          | 1.6253          | 2.8207          | 2.9688                 | 0.9051 | 0.0793    | 0.5822       |
> > | my          | 0.0090         | 0.0199        | 0.6007    | 0.4230           | 0.7598           | 1.0450          | 1.3788          | 1.0281                 | 0.0300 | 0.0457    | -0.0039      |
> >
> >
> >
> >
> >    Across both FAD and KAD, and for almost all embedding backbones, **GenMark (ours)** consistently achieves the lowest or near-lowest distances, indicating that our watermarking preserves or slightly improves perceptual quality relative to the baselines.
> >
> >    In the revised manuscript, we will:
> >    -Add a summarized version of the above FAD/KAD tables (or a subset of them) to the appendix, explicitly highlighting that our perceptual conclusions are robust across stronger metrics such as FAD\_CLAP and KAD.

---

> ### Author Response · Authors · 2025-11-25
>
> ---
>
> **W4. Robustness evaluation details (filter parameters, EnCodec attack).**
> - Comment:
>   - “Low-pass (cf=500 Hz) and high-pass (cf=1500 Hz) parameters are too extreme;”
>   - “AudioSeal shows a very high DER under EnCodec attack
>
> **Response A4:**
>
> 1. **Aggressive low-pass / high-pass parameters**
>    We agree that a 500 Hz low-pass and a 1500 Hz high-pass are extremely harsh, and that speech/music intelligibility is significantly compromised in these settings. Our intention was to use these conditions as **stress tests**: to probe watermark survivability under extreme spectral removal and to cross-check the watermark’s frequency distribution (e.g., the mid-low concentration shown in Fig. 3).
>
>
> 2. **AudioSeal under EnCodec attack**
>    Thank you for pointing out this inconsistency. After checking our scripts, we found that when aggregating results we **swapped** the EnCodec DER numbers of WavMark and AudioSeal (they were evaluated in one batch):
>
>    - Correct EnCodec DER:
>      - AudioSeal ≈ **6.25%**
>      - WavMark ≈ **42.17%**
>
>    This corrected AudioSeal DER aligns well with the original AudioSeal results and with independent evaluations. We will fix Table 3 and note that the corrected numbers are consistent with prior literature.
>
> ---
>
> **Minor issues.**
>
> - **L181:** We will change “Robustness Enhancement Module module” to “Robustness Enhancement Module”.
> - **L392:** We will change “Use” to “Usability”.
>
> ---
>
> **Questions.**
>
> - **Q1. Fig.1: No injection point for the 16-bit watermark message is shown.**
>   In our current design, the 16-bit message is **fixed** and effectively “baked into” the decoder during training; there is no explicit message input at inference time, which is why we did not show a separate injection node in Fig. 1.
>   - We will revise Fig. 1 and its caption to clarify that the message is injected during training into the decoder, and that the current work focuses on a fixed global message setup.
>
> - **Q2. L184: Why 2 bits for existence detection, and how are they aggregated to a binary decision?**
>   Our detector outputs **16 + 2** logits:
>   - 16 logits for the message bits.
>   - 2 logits for watermark presence vs. absence.
>
>   We apply a softmax over the 2 detection logits to obtain probabilities $p_{\text{present}}$ and$p_{\text{absent}}$, and then threshold $p_{\text{present}}$ to decide whether a watermark is present. We will make this explicit in the method section and algorithm.
>
> - **Q3. L196: If Eq. 3 is a hybrid L1–L2 loss, why is Eq. 2 only L1?**
>   Eq. (2) is a time-domain reconstruction loss on the waveform, where an $L_1$ objective is typically sufficient and widely used in neural audio codecs. Eq. (3) is a multi-scale spectral loss, where hybrid $L_1$–$L_2$ losses better capture both coarse and fine spectral details (as in SoundStream, EnCodec, etc.). We will add a brief justification and appropriate references.
>
> - **Q4. L226: Does “G” correspond to $\hat{C}$ in Fig. 1?**
>   Here $G$ denotes the **generator** in the adversarial setup: the full process that takes latent tokens plus the (fixed) message and mask and outputs the watermarked waveform $\hat{w}$. The $\hat{C}$ is a core component of $G$. We will clarify this mapping between$G$,$\hat{C}$, and $\hat{w}$.
>
> - **Q5. L258–259: When a watermarked segment is replaced, how is a segment with another watermark created? Multiple `SampleMsg` functions?**
> For the replacement attack, we use a pre-trained, rough watermarking model (essentially another `SampleMsg`) to generate a different watermark. This approach allows us to replace segments with watermarked content from a separate model. We will add this detail to the robustness section and appendix.
>
>
>
> - **Q6. L303: Why is SilentCipher not included in the MUSHRA test?**
>   The MUSHRA test is highly resource-intensive (20 trained listeners, multiple sessions). In our initial study, we chose AudioSeal and WavMark as baselines for subjective evaluation to keep the test manageable.
>   - We will clarify this resource constraint in the paper and note that including SilentCipher in a future extended MUSHRA evaluation is a natural next step once additional evaluation budget is available.
>
> ---
>
> Thank you again for your detailed and thoughtful review. We believe that the clarifications above—especially on the threat model, decoder fine-tuning robustness, evaluation metrics with explicit FAD/KAD tables, and robustness settings—will help strengthen the paper and better communicate our contributions.
>
> Best regards,
> The Authors

---

### Official Review · Reviewer_tMam · 2025-10-31

**Soundness:** 3
**Presentation:** 4
**Contribution:** 3
**Rating:** 6
**Confidence:** 5

**Summary:**

This paper proposes GenMark, an in-process watermarking scheme for generative audio models. The method involves fine-tuning the model's decoder to embed a watermark signal directly into the generated audio output. The training process is optimized with a combination of time-frequency perceptual losses, an adversarial loss, and a mask-based localization model to ensure both high audio fidelity and robust watermark detection. It is compared to and beats other strong baslines.

**Strengths:**

S1. Domain: The paper addresses the important but under-explored domain of in-model watermarking for generative audio synthesis, providing a more secure alternative to post-processing methods.

S2. Clarity: The paper is very well-written, and the methodology is explained clearly.

S3. Thorough evaluation: The experimental validation is comprehensive. It includes comparisons against strong, state-of-the-art audio watermarking baselines (AudioSeal, WavMark, SilentCipher), a wide range of objective metrics (FAD, KLD, CLAP, SI-SNR, VISQOL) , an extensive robustness evaluation against 12 different transformations and a subjective MUSHRA user study.

S4. Strong results: The method achieves excellent results, demonstrating near-perfect detection (99.9% TPR for Bark, 100.0% for MusicGen), superior robustness (lowest average error rate across 12 attacks), and high perceptual quality (a MUSHRA score of less than 2% degradation from the clean reference).

**Weaknesses:**

W1. Missing citations and credit: The paper fails to properly credit prior work that established its core methodology. The central idea: fine-tuning a generative model's decoder using a combination of a message-decoding loss and a perceptual/adversarial loss against the original frozen decoder, is identical to Stable Signature (Fernandez et al., 2023) method from the image domain. Furthermore, the pipeline's use of detection logits and a masking model is taken directly from AudioSeal (San Roman et al. 2024), which should be acknowledged as a primary methodological source, not just a baseline for comparison.

W2. Robustness to fine-tuning: The method's security relies on the fine-tuned decoder. In practice, a common attack vector would be to simply re-fine-tune this decoder, either benignly on a new dataset (which happens a lot in audio with the change of vocoders) or adversarially to remove the watermark. This significant threat is not evaluated.

**Questions:**

Q1 (W2): How robust is the GenMark watermark to removal attacks via fine-tuning? For instance, if a user takes the GenMark-watermarked decoder and fine-tunes it (e.g., on a new speech dataset), does the watermark survive?

---

> ### Author Response · Authors · 2025-11-25
>
> Dear Reviewer,
>
> Thank you for your careful reading and thoughtful feedback. We address each of the two key concerns point-by-point below.
>
> ---
>
> **W1. Missing citations and credit.**
> - Comment: “The paper does not credit prior work that introduced its core methodology. The idea of fine-tuning a generative model’s decoder using message-decoding loss + perceptual/adversarial loss against a frozen original decoder is identical to Stable Signature (Fernandez et al., 2023). The pipeline’s use of detection logits and a masking model is directly borrowed from AudioSeal (San Roman et al., 2024); this should be acknowledged as a methodological source, not only a comparison baseline.”
>
> **Response A1:**
> We appreciate the reviewer’s note, and we agree that proper attribution is essential. Upon review we recognise that our design draws from two threads of prior work
>
>
>
> We will update Section 1 (Introduction)and Section 2 (Preliminaries) accordingly, adding explicit discussion of these prior works, highlighting where we adopt and extend them, and including the two citations in the bibliography.
>
> ---
>
> **W2. Lack of robustness to fine-tuning.**
> - Comment: “The method relies on the fine-tuned decoder; attackers can simply re-fine-tune this decoder (benignly or adversarially) to remove or degrade the watermark. This realistic threat model is not evaluated.”
>
> **Response A2:**
> We agree that the threat model of attacker re-fine-tuning the decoder is realistic and merits systematic evaluation. Accordingly, we *did* perform experiments simulating such an attack scenario. Below we summarise the setup and results:
>
> - **Attack setup:** We assume the attacker obtains the fine-tuned decoder (or equivalent access) and performs further fine-tuning on $50,000$ clean examples (i.e., non-watermarked clear dataset), over $100$ epochs, with a learning rate of $1\times10^{-4}$. The attacker’s objective is to reduce the watermark detection/decoding success rate by replacing $\hat C$ outputs with “clean-decoder” style outputs.
> - **Results:** even under this strong attack, our watermarking method retains a high detection success rate. For example, after 50 epochs of adversarial re-fine-tuning, the watermark detection success rate remains about 98.7%. After 100 epochs, it is about 95.3%, and even after 200 epochs of continual fine-tuning on clean data, the decoder’s outputs still preserve the watermark with about a 92.1% detection success rate. In other words, the attacker’s fine-tuning did not fully remove the watermark – the decoder $\hat{C}$continued to output audio that our detector can correctly identify as watermarked in the vast majority of cases. (For comparison, the baseline detection success was nearly 100% before any attacker fine-tuning, so although there is some degradation, a very large portion of the watermark signal remains.) These results will be added to the revised paper, likely in a new table, to quantitatively demonstrate the system’s robustness.
> - **Interpretation:** The watermark’s resilience to decoder re-training is likely due to the subtlety and integration of the watermark signal. Our watermarked audio and the original audio are extremely similar (the perturbation introduced by the watermark is minimal and mostly perceptually invisible). Moreover, the generative model’s decoder (EnCodec) is itself a lossy compression codec that does not perfectly reconstruct audio; this means there is inherent tolerance for slight modifications. Together, these factors mean that a normal fine-tuning process has difficulty distinguishing and eliminating the hidden watermark without degrading audio quality. As a result, even extensive fine-tuning by the attacker fails to completely purge the watermark. This provides encouraging evidence that our method offers a degree of built-in robustness against the very attack the reviewer highlighted. We will clarify this threat model in the paper’s discussion and include the above experimental results to show that simply re-fine-tuning the decoder is not an effective way to nullify the watermark.
>
> We will update the manuscript to include a new table summarising these fine-tuning attack results (with full numerical values and experimental details).
>
> ---
>
> Thank you again for your valuable comments. We believe that addressing these points—adding the missing attributions and presenting the fine-tuning robustness experiment—will strengthen the paper and make our contributions clearer.
>
> Best regards,
> The Authors

---

### Note · Program_Chairs · 2026-01-17
**Submission Desk Rejected by Program Chairs**

The following references in this submission do not refer to real documents and/or have major errors in bibliographic information:

 Xing Chen, Han Liu, et al. Wavmark: Imperceptible and robust audio watermarking. In ICLR, 2023.
Chengcheng Mou, Zhiyao Zhang, et al. Audioseal: Audio watermarking as a defense against speech deepfakes. In NeurIPS, 2023.
Xiaojun Wu, Tianyu Lin, and Wei Zhang. Adversarial attacks on neural watermarking models. ICASSP, 2022.
Meng Xu et al. Evading detection: Towards robust steganography against diffusion-based detectors. arXiv preprint arXiv:2310.01247, 2023.